



# Profiling the Molecular Destruction Rates of Temperature and Humidity as well as the Turbulent Kinetic Energy Dissipation in the Convective Boundary Layer

Volker Wulfmeyer[1], Christoph Senff[2,3], Florian Späth[1], Andreas Behrendt[1], Diego Lange[1], Robert M. Banta[2,3], W. Alan Brewer[3], Andreas Wieser[4], and David D. Turner[5]

[1]Institute of Physics and Meteorology, University of Hohenheim, Stuttgart, Germany
[2]Cooperative Institute for Research in Environmental Sciences (CIRES), University of Colorado, Boulder, Colorado, USA
[3]NOAA Chemical Sciences Laboratory (CSL), Boulder, Colorado, USA
[4]Karlsruhe Institute of Technology, Karlsruhe, Germany
[5]NOAA Global Systems Laboratory (GSL), Boulder, Colorado, USA

**Correspondence:** Volker Wulfmeyer (volker.wulfmeyer@uni-hohenheim.de)

**Abstract.** A simultaneous deployment of Doppler, temperature, and water-vapor lidars is able to provide profiles of molecular destruction rates and turbulent kinetic energy (TKE) dissipation in the convective boundary layer (CBL). Horizontal wind profiles and profiles of vertical wind, temperature, and moisture fluctuations are combined for deriving the dissipation and molecular destruction rates by determining the transversal temporal autocovariance functions (ACFs). These ACFs are fitted

to their theoretical shapes and coefficients in the inertial subrange. Error bars are estimated by a propagation of noise errors. Sophisticated analyses of the ACFs are performed in order to choose the correct range of lags of the fits for fitting their theoretical shapes in the inertial subrange as well as for minimizing systematic errors due to temporal and spatial averaging and micro- and mesoscale circulations. We demonstrate that we achieve very consistent results of the derived profiles of turbulent variables regardless whether 1-s or 10-s time resolutions are used. We also show that the temporal and spatial length

scales of the fluctuations of vertical wind, moisture, and potential temperature are similar with a spatial integral scale of $\approx 160\,\mathrm{m}$ at least in the mixed layer (ML). The profiles of the molecular destruction rates show a maximum in the interfacial layer (IL) and reach values of $\epsilon_m \simeq 7 \cdot 10^{-4}\mathrm{g}^2\mathrm{kg}^{-2}\mathrm{s}^{-1}$ for mixing ratio and $\epsilon_\theta \simeq 1.6 \cdot 10^{-3}\mathrm{K}^2\mathrm{s}^{-1}$ or potential temperature. In contrast, the maximum of the TKE dissipation is reached in the ML and amounts $\simeq 10^{-2}\mathrm{m}^2\mathrm{s}^{-3}$. We also demonstrate that the vertical wind ACF coefficient $k_w \propto \overline{w'^2}$ and the TKE dissipation $\epsilon \propto \left(\overline{w'^2}\right)^{3/2}$. For the molecular destruction rates we show

that $\epsilon_m \propto \overline{m'^2} \left(\overline{w'^2}\right)^{1/2}$ and $\epsilon_\theta \propto \overline{\theta'^2} \left(\overline{w'^2}\right)^{1/2}$. These equations can be used for parameterizations of $\epsilon$, $\epsilon_m$, and $\epsilon_\theta$. All noise errors bars are derived by error propagation and are small enough to compare the results with previous observations and large eddy simulations. The results agree well with previous observations but show more detailed structures in the IL. Consequently, the synergy resulting from this new combination of active remote sensors enables the profiling of turbulent variables such as integral scales, variances, TKE dissipation, and the molecular destruction rates as well as deriving relationships between them.

The results can be used for the parameterization of turbulent variables, TKE budget analyses, and the verification of large eddy simulations.



# 1 Introduction

Improved understanding and modeling of turbulent transport processes in the convective boundary layer (CBL) requires in-depth studies of the budgets of second and other higher-order moments of atmospheric variables. Key variables include turbulent kinetic energy (TKE) and water-vapor and temperature variances; the latter in terms of absolute humidity, mixing ratio, or specific humidity variances as well as of temperature or potential temperature variances. The analysis of TKE and variance budgets and their components is important for the verification of weather forecast, climate, and earth system models. Furthermore, the representation of budgets of second-order moments of atmospheric variables is essential for the parameterization of turbulent transport processes in mesoscale models, in which the turbulence is not explicitly resolved, or for the parameterization of sub-grid scale cloud processes.

In most turbulence parameterizations (TP), the variances of scalars are not considered. However, this is not the case in higher-order closure schemes, which require the modeling and simulation of the scalar variance budget equations (see, e.g., Larson and Golaz (2005); Bogenschutz et al. (2012)). Starting points for the analyses are the TKE and variance budget equations. A component of the governing equation for the TKE is the TKE dissipation term. This rate must be parameterized in mesoscale models. Various parameterizations of the TKE dissipation have been proposed such as in Nakanishi and Niino (2009) for the MYNN TP or the combined MYNN-EDMF scheme (Olson et al., 2019). Also the destruction rates of temperature and moisture variances must be considered and parameterized. Another example where the study of variance budgets is required is the parameterization of sub-grid clouds in mesoscale models (Golaz et al., 2002; Van Weverberg et al., 2016).

Large eddy and direct numerical simulation (LES and DNS) models predict turbulence fields by explicitly resolving turbulent processes into the inertial subrange. Resolving these processes avoids the need for TPs at their spatial grid increments ($\approx< 100\,\mathrm{m}$), but it is still necessary to parameterize the remaining sub-grid scale processes such as the TKE dissipation and molecular destruction rates.

Therefore, from the meso- to the turbulent scales, vertical profiling of TKE dissipation and destruction rates of molecular variances are very useful. On the mesoscale, these measurements can be used for the verification of TPs. On the smaller scales, these observations are very important for the verification of the performance of LES and DNS.

This verification should be performed under a range of different meteorological conditions, from the surface layer (SL), through the mixed layer (ML) and the interfacial layer (IL) at the CBL top to the lower troposphere. Previous studies, as early as the 1970s, used mainly in-situ measurements (Caughey and Palmer, 1979; Lenschow et al., 1980). These sensors were operated on research aircraft; in the future, it should also be possible to perform these measurements on unmanned aerial vehicles (UAVs). However, turbulence measurements from aircraft and UAVs are limited with respect to duration and vertical range, and there can be questions as to where the observations are located relative to the CBL top (Turner et al., 2014a). Furthermore, it is difficult to measure instantaneous vertical profiles of mean and turbulent variables, thereby limiting their applications for process studies and model verification.

A powerful alternative is the operation and application of ground-based active remote systems which are considered in this work. In the SL, scanning lidar system were successfully combined to derive high-resolution profiles for studying Monin-



Obukhov similarity theory (Späth et al., 2022) but close to the surface, the temporal-spatial resolutions of lidar systems is not sufficient to resolve the major part of the turbulent fluctuations. However, due to some recent technological advances, Doppler lidar (DL) systems are now available for 24/7 measurements of horizontal and vertical wind profiles in the CBL. The range and time resolutions of line-of-sight (LOS) velocity measurements with DLs is high enough to resolve turbulent processes

above the SL from the ML to the IL. Recently, it has been demonstrated that conically scanning DLs are capable of measuring horizontal wind profiles with high vertical and temporal resolution as well as accuracy (Berg et al., 2017). Six-beam staring modes of DLs have been developed with even greater information content, as TKE, momentum flux, and horizontal wind profiles can be determined simultaneously (Bonin et al., 2017). Furthermore, vertically staring DLs have been extensively used for the measurement of vertical wind ($w$) statistics (Lenschow et al., 2000; Wulfmeyer and Janjić, 2005; Hogan et al., 2009;

Lothon et al., 2009; Tucker et al., 2009; Ansmann et al., 2010; Lenschow et al., 2012) and TKE dissipation profiles (Frehlich and Cornman, 2002; O'Connor et al., 2010; Banakh et al., 2017; Bodini et al., 2018; Wildmann et al., 2019).

For the measurement of scalar variances and their destruction rates, high-resolution vertical profiling of temperature and moisture is also possible with active remote sensing. Sufficient spatial-temporal resolutions can be reached with water-vapor differential absorption lidar (WVDIAL) and the water-vapor Raman lidar (WVRL) techniques. For WVDIAL, this was demon-

strated in Wulfmeyer (1999); Lenschow et al. (2000); Späth et al. (2016); Muppa et al. (2016); Wulfmeyer et al. (2016). For WVRL, this performance was confirmed in Wulfmeyer et al. (2010); Turner et al. (2014a, b); Wulfmeyer et al. (2018); Osman et al. (2019).

With respect to temperature measurements, recently a breakthrough has been achieved using the temperature rotational Raman lidar (TRRL) technique. The first demonstration of high-resolution temperature profiling during daytime in the CBL,

from which higher order moments could be derived, was presented in Hammann et al. (2015); Behrendt et al. (2015), and the first combined measurements of sensible and latent heat flux profiles in Behrendt et al. (2020). Furthermore, Lange et al. (2019) showed that even quasi-operational measurements down to a vertical resolution of 7.5 m and a temporal resolution of 10 s are possible so that it is straightforward to analyze temperature variance profiles in the daytime CBL. Therefore, the combination of DL, WVRL/WVDIAL, and TRRL can be used for extensive turbulence studies based on their single profiles

or a combination of these. Further details about the combined WV and T Raman lidar methodology are found in Wulfmeyer and Behrendt (2021).

DL measurements are either available from observatories such as from the Atmospheric Radiation Measurement (ARM) Program Southern Great Plains (SGP) site (Sisterson et al., 2016), where DLs have been operated for several years in an alternating conically scanning and vertical pointing mode (Berg et al., 2017), the Land-Atmosphere Feedback Observatory (LAFO,

see https://lafo.uni-hohenheim.de/en, Späth et al. (2023)), as well as other observatories. However, combined measurements with coincident WVDIAL, WVRL, and TRRL are sparse. Currently, to our knowledge the longest, operational dataset that has simultaneous, high-resolution DL and WVRL observations was collected at the SGP site. Further dedicated data sets were collected during various field campaigns such as the High Definition Clouds and Precipitation (HD(CP)$^2$) Observational Prototype Experiment (HOPE) (https://hdcp2.zmaw.de) (Macke et al., 2017) and the Land-Atmosphere Feedback Experiment (LAFE)

(Wulfmeyer et al., 2018).





In this work, we present the derivation of TKE dissipation as well as molecular destruction rate profiles from the HOPE campaign. This study is organized as follows: In section 2, we present the field campaign and the data set used in this study. We revisit the TKE, water-vapor, and temperature budget equations and discuss the terms containing the dissipation rates in section 3. Some examples of their parameterizations are presented. In section 4, we show how we derived and evaluated the transverse temporal autocovariance functions (ACFs) and the power spectra of the lidar time series in order to derive profiles of variances, the ACF coefficients, and the integral time scales in dependence of temporal and spatial resolutions. First examples of the profiles of TKE dissipation as well as of molecular destruction rates using our method for a case during HOPE are presented in section 5. These results contain detailed error analyses. In section 6, the results are discussed and compared with previous methods and corresponding results. A summary and an outlook are given in section 7.

## 2 HOPE campaign: Description of the case and the data sets

We present analyses of turbulence profiles from Intensive Observations Period (IOP) 5 of HOPE, which was performed in Spring 2013 close to the city of Jülich in Germany. IOP5 was executed on 20 April 2013. The data set was collected with the WVDIAL and the TRRL of the Institute of Physics and Meteorology (IPM) at the University of Hohenheim (UHOH) as well as a DL operated by the Karlsruhe Institute of Technology (KIT) between 11:30-12:30 UTC. The lidar systems were located at site close to the village of Hambach near Research Centre Jülich at 50° 53' 50.56" N and 6° 27' 50.39" E, 110 m above sea level. During this time period, the atmosphere was cloud free and contained only a few aerosol layers in the free troposphere. Around local noon, the surface sensible heat flux was $\approx 250\,\mathrm{W\,m^{-2}}$, which corresponds to a kinematic heat flux of $0.2\,\mathrm{K\,m\,s^{-1}}$ whereas the latent heat flux was $\approx 90\,\mathrm{W\,m^{-2}}$. The Obukhov length was $L_0 = -126\,\mathrm{m}$ and the convective velocity scale $w_* \simeq 0.7\,\mathrm{m\,s^{-1}}$ leading to a quasi-steady CBL depth with $z_i \approx 1280\,\mathrm{m}$. Consequently, according to Lothon et al. (2006), we were dealing with a weak convective case as $\zeta = -z_i/L_0 \simeq 10$ was small. Further details of the data sets and the meteorological conditions are presented in Wulfmeyer et al. (2016); Muppa et al. (2016).

The UHOH WVDIAL is based on a Ti:Sapphire laser transmitter tuned to $\approx 820\,\mathrm{nm}$, which can be operated up to an average power of 10 W, in combination with a very efficient receiver. The transmitter fulfills all requirements for very accurate absolute humidity measurements due to its excellent stability and the narrow bandwidth of the laser spectrum (Wagner et al., 2013). The receiver consists of an 80-cm, 3D scanning telescope, a high-transmission, narrow-band interference filter, and an Avalanche photodiode. This system permits the measurement of water-vapor profiles with a temporal resolution of 1-10 s and a spatial resolution of 15-150 m in the lower troposphere. Further details are found in Späth et al. (2016).

The UHOH TRRL is a combined water-vapor and temperature rotational Raman lidar. The laser transmitter is a frequency-tripled, injection-seeded Nd:YAG laser, which delivers up to 15 W average power at 355 nm. The heart of this system is a very efficient, high-transmission series of interference filters in front of four sensitive photomultipliers, which collect four signals: the elastic backscatter, two channels sensitive to the rotational Raman scattering by nitrogen and oxygen, and the vibrational-rotational Raman channel of water vapor. In this work, we focus on the temperature profiles measured with the





TRRL. Hammann et al. (2015) and Behrendt et al. (2015, 2020) showed that this system is capable of daytime temperature profiling with a temporal resolution of 10 s and a spatial resolution of 30-150 m in the lower troposphere.

At the Hambach site, KIT operated a coherent DL based on a Er:YAG, 1.6-$\mu$m laser transmitter (Wind-Tracer "WTX" from Lockheed Martin Coherent Technologies, Inc.). This DL measured the line-of sight wind velocity with a temporal resolution of 1-10 s and an effective range-resolution of $\approx 60$ m (Träumner et al., 2011). The DL measurements covered the CBL up to the top of the boundary layer and partly above, depending on the aerosol particle concentration. Mean horizontal wind and vertical wind profiles were determined by alternating between a velocity-azimuth display (VAD) algorithm and a vertically

staring mode (Maurer et al., 2016). The DL was running in the vertical mode for 56 min and then switched to a scanning mode with conical scans (PPIs) at elevation angles of 5 and 75 degrees with a rotation speed of 6 degrees per second followed by 3 slice scans (RHI) at azimuth positions shifted about 120 degrees. Thus, an averaging time of $\approx 240$ s was used for deriving a horizontal wind profile. Centered in the time period used for the turbulence profiling, this scan was performed between 12:00:10-12:04:30 UTC. Due to the high SNR of the KIT DL in the CBL, vertical wind profiles can be determined with

resolutions of 1 s and 50 m, respectively.

    The WVDIAL, TRRL, and the DL data were processed consistently with a temporal resolution of either 1 s or 10 s with the same time steps from 11:30-12:30 UTC. All measurements started at 350 m above ground level (agl). We maintained a vertical resolution of 50 m in the DL vertical wind profiles, as the resulting signal-to-noise-ratio (SNR) was good enough for accurate measurements up to the IL. The WVDIAL and the TRRL data were processed with vertical resolutions of 70 m and

100 m, respectively, in order to maintain an acceptable SNR up to the IL. For the WVDIAL, we evaluated data with 1 s and 10 s resolutions for studying changes in the ACFs and the power spectra whereas the TRRL data could only be evaluated with 10 s due to the lower SNR. The TRRL data were overlap corrected up to 800 m (Hammann et al., 2015) and the WVDIAL data up to 300 m with constant correction functions. Afterwards, the WVDIAL absolute humidity measurements and the TRRL temperature measurements were transformed into specific humidity $s$ or mixing ratio $m$ as well as potential temperature $\theta$ using

a hydrostatic pressure profile. The analysis could be performed using either $s$ or $m$; however, as the absolute humidity was only a few g m$^{-3}$ the relative difference between $m$ and $s$ was in the sub-percentage region. We preferred to derive and analyze profiles of $m$ and $\theta$ for the comparison with previous measurements as well as with LES and mesoscale model output, as these are the typical prognostic variables. Finally, all data were gridded to a vertical range grid of 15 m from 350 m to 1600 m AGL. Also, radiosonde profiles at 11 UTC and 13 UTC were included in the horizontal wind analyses in order to close measurement

gaps of the DL in the IL.

## 3   Vertical profiles of TKE dissipation as well as molecular destruction rates

### 3.1   Governing equations and parameterizations

The TKE dissipation and the molecular destruction rates are the sink or loss terms in the TKE and mixing ratio and potential temperature variance budget equations. TKE is defined as $e = (u'^2 + v'^2 + w'^2)/2$ where $u'$, $v'$, and $w'$ are the fluctuations of





the three wind components. Under horizontally homogeneous conditions, the TKE prognostic equation reads

$$\frac{\partial e}{\partial t} \simeq -\frac{\partial}{\partial z}\left[\overline{w'\left(e+\frac{p'}{\rho_0}\right)}\right] - \overline{u'w'}\frac{\partial u}{\partial z} - \overline{v'w'}\frac{\partial v}{\partial z} + \frac{g}{\theta_0}\overline{w'\theta'_v} - \epsilon \tag{1}$$

where $\epsilon$ is the TKE dissipation, $p'$ is the atmospheric pressure fluctuation, $\rho_0$ is air density, $\overline{u'w'}$ and $\overline{v'w'}$ are the momentum fluxes, $g$ is the acceleration due to gravity, $\theta_0$ is the potential temperature, and $\overline{w'\theta'_v}$ represents the buoyancy flux. In this version of Eq. 1, the horizontal gradients of fluxes are neglected.

The TKE dissipation can be parameterized in multiple ways; see, e.g., Deardorff (1973); Mellor and Yamada (1974). For instance, Nakanishi and Niino (2009) represent it as

$$\epsilon \approx \frac{e^{3/2}}{B_1 l} \tag{2}$$

where $B_1$ is a closure coefficient and $l$ is the mixing length or grid scale. Of course, this is only one example; there are many other parameterizations available. In any case, if $\epsilon$ and TKE can be measured simultaneously, it is possible to evaluate this 165 parameterization.

The budgets for water-vapor mixing ratio variance $\overline{m'^2}$ and the potential temperature variance $\overline{\theta'^2}$, respectively, are given by

$$\frac{\partial \overline{m'^2}}{\partial t} \simeq -2\overline{w'm'}\frac{\partial m}{\partial z} - \frac{\partial}{\partial z}\overline{w'm'^2} - 2\epsilon_m \tag{3}$$

$$\frac{\partial \overline{\theta'^2}}{\partial t} \simeq -2\overline{w'\theta'}\frac{\partial \theta}{\partial z} - \frac{\partial}{\partial z}\overline{w'\theta'^2} - 2\epsilon_\theta \tag{4}$$

also under horizontally homogeneous conditions. Here, $\epsilon_m$ and $\epsilon_\theta$ are the molecular destruction rates of $m$ and $\theta$, respectively. 170 Our objective here is to directly measure profiles of $\epsilon_m$ and $\epsilon_\theta$.

As mentioned above, in standard TPs, parameterizations of the variance budget equations are not necessary, except when higher-order turbulence closures are used. Several groups are starting to evaluate the use of these advanced closures schemes such as in MYNN now. In LES and DNS, the molecular destruction rates are not parameterized but it is assumed that these are resolved or negligible. Thus, a comparison of their simulations and our measurements can be used to study the sub-grid scale 175 closure of these models.

### 3.2 Derivation of dissipation and destruction rate profiles

Wulfmeyer et al. (2016) introduced a method to measure vertical profiles of $\epsilon$, $\epsilon_m$, and $\epsilon_\theta$. For the sake of completeness, we summarize here the most important steps. We start with the derivation of the structure function $D(r)$ and its transformation in the temporal domain using Taylor's hypothesis (see also Tatarski (1961) and Monin and Yaglom (1975)). The relation between 180 the structure function and the ACF is presented and discussed more in detail in Wulfmeyer et al. (2016).

If the turbulence is stationary and isotropic and the temporal and vertical resolutions of the lidar observations are high enough to resolve the inertial subrange, then the resulting transversal temporal ACFs $A_i(\tau)$ with lag $\tau$ of the fluctuations of vertical wind $w'$, mixing ratio $m'$, and potential temperature $\theta'$ at each height range $z$ show a dependence of $\tau^{2/3}$ and can be written



as follows:

$$A_{w'}(\tau) = \overline{w'^2} - k_w \tau^{2/3} = \overline{w'^2} - \epsilon^{2/3} V^{2/3} \tau^{2/3} \tag{5}$$

$$A_{m'}(\tau) = \overline{m'^2} - k_m \tau^{2/3} = \overline{m'^2} - 0.5 a_m^2 \frac{\epsilon_m}{\epsilon^{1/3}} V^{2/3} \tau^{2/3} \tag{6}$$

$$A_{\theta'}(\tau) = \overline{\theta'^2} - k_\theta \tau^{2/3} = \overline{\theta'^2} - 0.5 a_\theta^2 \frac{\epsilon_\theta}{\epsilon^{1/3}} V^{2/3} \tau^{2/3} \tag{7}$$

where the constants $a_m^2$ and $a_\theta^2$ are expected to be in the range of $2.8 - 3.2$ (Stull, 1988). The determination of the variances and the ACF coefficients $k_i$ can be achieved by the procedures introduced in Lenschow et al. (2000) and Wulfmeyer et al. (2016). As the lidar measurements are performed in the temporal domain, the horizontal wind profile $V$ must also be determined to derive the dissipation and the molecular destruction rates.

Generally, lidar-based measurements of the ACFs are influenced by system noise and spatial and temporal filtering effects. Therefore, it is reasonable to study the ACFs using the following transformations:

$$\ln\left(\overline{w'^2} - A_{w'}(\tau)\right) = \frac{2}{3}\ln(\tau) + \ln(k_w) \tag{8}$$

$$\ln\left(\overline{m'^2} - A_{m'}(\tau)\right) = \frac{2}{3}\ln(\tau) + \ln(k_m) \tag{9}$$

$$\ln\left(\overline{\theta'^2} - A_{\theta'}(\tau)\right) = \frac{2}{3}\ln(\tau) + \ln(k_\theta) \tag{10}$$

Alternatively, the transformations

$$k_w = \frac{\overline{w'^2} - A_{w'}(\tau)}{\tau^{2/3}} \tag{11}$$

$$k_m = \frac{\overline{m'^2} - A_{m'}(\tau)}{\tau^{2/3}} \tag{12}$$

$$k_\theta = \frac{\overline{\theta'^2} - A_{\theta'}(\tau)}{\tau^{2/3}} \tag{13}$$

can be used to identify the most reasonable range of the number of lags to use to derive the fit (we will refer to this number as the "fitlags") because in this region the ACF coefficients should be constant. Consequently, these Eqs. 8-10 and 11-13 can be applied for lags $\geq 1$ to investigate whether the lidar measurements agree with the theoretical shape of the ACFs and in what range of lags the ACF coefficients should be derived in dependence of the time resolution (see below section 4).

The ACFs should also be studied with respect to their temporal and spatial integral scales $\mathcal{T}_w$ and $\mathcal{R}_w$ for $w'$, $\mathcal{T}_m$ and $\mathcal{R}_m$ for $m'$, and $\mathcal{T}_\theta$ and $\mathcal{R}_\theta$ for $\theta'$. This can be realized using the following relationships:

$$\mathcal{T}_w = \frac{2}{5}\left(\sqrt{\overline{w'^2}}\right)^3 \frac{1}{\epsilon V}, \qquad \mathcal{R}_w = \frac{2}{5}\left(\sqrt{\overline{w'^2}}\right)^3 \frac{1}{\epsilon} \tag{14}$$

$$\mathcal{T}_m = \frac{2\sqrt{8}}{5 a_m^3}\left(\sqrt{\overline{m'^2}}\right)^3 \sqrt{\frac{\epsilon}{\epsilon_m^3}} \frac{1}{V}, \qquad \mathcal{R}_m = \frac{2\sqrt{8}}{5 a_m^3}\left(\sqrt{\overline{m'^2}}\right)^3 \sqrt{\frac{\epsilon}{\epsilon_m^3}} \tag{15}$$

$$\mathcal{T}_\theta = \frac{2\sqrt{8}}{5 a_\theta^3}\left(\sqrt{\overline{\theta'^2}}\right)^3 \sqrt{\frac{\epsilon}{\epsilon_\theta^3}} \frac{1}{V}, \qquad \mathcal{R}_\theta = \frac{2\sqrt{8}}{5 a_\theta^3}\left(\sqrt{\overline{\theta'^2}}\right)^3 \sqrt{\frac{\epsilon}{\epsilon_\theta^3}} \tag{16}$$

Please note that the integral time scales can be directly derived from the ACFs using the equation (Wulfmeyer et al., 2016)

$$\mathcal{T}_i = \frac{2}{5}\left(\frac{var_{atm,i}}{k_i}\right)^{3/2} \tag{17}$$





where $\mathcal{T}_i$ is the integral time scale of the atmospheric variable of interest $i$, $var_{atm,i}$ is its atmospheric variance, and $k_i$ is the corresponding ACF coefficient (see Eqs. 5-7). This equation turned out to be most robust (i.e., least sensitive) to system noise. The variances can be measured with DL, WVDIAL, WVRL, TRRL, and the horizontal wind profiles can be measured with conically scanning DL or 6-beam staring DL. Obviously, this combination of instruments and measurement configurations is necessary but also complete for deriving the TKE dissipation and molecular destruction rates.

Solving Eqs. 5-7 or 14-16 for these variables yields for $\epsilon$

$$\epsilon = \frac{k_w^{3/2}}{V} \tag{18}$$

$$\text{or} \quad \epsilon = \frac{2}{5} \frac{\left(\sqrt{\overline{w'^2}}\right)^3}{V\,\mathcal{T}_w} = \frac{2}{5} \frac{\left(\sqrt{\overline{w'^2}}\right)^3}{\mathcal{R}_w}, \tag{19}$$

for $\epsilon_m$

$$\epsilon_m = \frac{2\,k_m\,\sqrt{k_w}}{a_m^2} \frac{1}{V} \tag{20}$$

$$\text{or} \quad \epsilon_m = \frac{4}{5} \frac{\overline{m'^2}\,\sqrt{\overline{w'^2}}}{a_m^2\,\mathcal{T}_m^{2/3}\,\mathcal{T}_w^{1/3}\,V} = \frac{4}{5} \frac{\overline{m'^2}\,\sqrt{\overline{w'^2}}}{a_m^2\,\mathcal{R}_m^{2/3}\,\mathcal{R}_w^{1/3}}, \tag{21}$$

and finally for $\epsilon_\theta$

$$\epsilon_\theta = \frac{2\,k_\theta\,\sqrt{k_w}}{a_\theta^2} \frac{1}{V} \tag{22}$$

$$\text{or} \quad \epsilon_\theta = \frac{4}{5} \frac{\overline{\theta'^2}\,\sqrt{\overline{w'^2}}}{a_\theta^2\,\mathcal{T}_\theta^{2/3}\,\mathcal{T}_w^{1/3}\,V} = \frac{4}{5} \frac{\overline{\theta'^2}\,\sqrt{\overline{w'^2}}}{a_\theta^2\,\mathcal{R}_\theta^{2/3}\,\mathcal{R}_w^{1/3}}. \tag{23}$$

Using Eqs. 20 and 22 we also achieve

$$\frac{k_m}{k_\theta} \simeq \frac{\epsilon_m}{\epsilon_\theta} \tag{24}$$

In this work, we apply Eqns. 18, 20, and 22 for the derivation of the dissipation and the molecular destruction rates, as it is the most direct way to derive and to use the required combination of structure coefficients.

## 4  Derivation of profiles of TKE dissipation and molecular destruction rates of water-vapor and temperature variances

### 4.1  Horizontal wind profile

As demonstrated in Eqns. 18-23, analyzing the turbulence profiles using the transversal ACFs or the power spectra requires that the horizontal wind profile be measured simultaneously. We applied and merged two data sources to get the horizontal wind profile, namely the horizontal wind profiles derived from conical scans of the KIT DL and radiosonde wind observations. We used the three hourly conical scans performed by the KIT DL between 11 UTC and 13 UTC and the radiosounding at 13 UTC to derive a best estimate of the horizontal wind profile during the period of interest.



**Figure 1.** Horizontal wind profiles determined from a radiosonde launch at 13 UTC and three conical DL scans. For the soundings (black bullets), an error of $1\,\mathrm{m\,s^{-1}}$ was assumed. The errors in the DL profiles were derived by error propagation of the uncertainty in radial velocities. The red bullets show the weighted average of these profiles, which was used for the derivation of turbulence profiles. The grey dashed line indicates $z_i$.





The results are presented in Fig. 1. The boundary layer depth $z_i \simeq 1280\,\mathrm{m}$ was determined using the WVDIAL backscatter lidar statistics and is indicated by the dashed gray line. The 5-min DL-derived wind profiles at 11, 12, and 13 UTC are shown
in green, blue, and pink, with error bars derived from DL radial velocity statistics. The sounding from 13 UTC is shown in black. For this profile, an error of $1\,\mathrm{m\,s^{-1}}$ at each height level was assumed. As the SNR of the wind measurements was not good enough to retrieve the wind profile in the IL above 1300 m, the information of the sounding was taken in this region. A best estimate of the horizontal wind profile during the turbulence measurement period was derived by using a noise error weighted average of the DL and the radiosonde profiles. The resulting wind profile and its error estimates are presented by
the red bullets and error bars. This wind profile and its error bars, gridded to 15 m in the vertical, were used for deriving the dissipation and molecular destruction profiles, as well as the uncertainties in these profiles using standard error propagation (see also Wulfmeyer et al. (2016), Appendix b.4). The largest deviation between the wind profiles occurs in the ML. However, the resulting weighted mean shows reasonable, nearly constant wind speed in the CBL and a windshear-driven increase of wind speed in the IL.

**4.2   Methodologies and procedures for the determination of variances and the ACF coefficients**

The determination of the ACFs and its coefficients with respect to its theoretical shape in the inertial subrange is more demanding than the estimation of variances for two reasons: 1) The difference of the total and the atmospheric variances around lag zero have to be determined with high accuracy. 2) A suitable range of lags for the ACFs must be evaluated for deriving its coefficient by a fit to the data. The determination of this coefficient is influenced by system noise, filter effects of temporal and
vertical averaging, and atmospheric effects, e.g., the influence of regional meso- or microscale circulations that may influence the fluctuations on the corresponding temporal and spectral scales.

For the derivation of turbulence profiles we applied Taylor's hypothesis of frozen turbulence. We assumed that this was valid, as $z_i$ was quasi steady over our analysis period, the CBL was nearly well-mixed, the temporal and vertical variability in the horizontal wind profiles $V(z)$ was rather small, and the mean horizontal wind speed in the CBL was high with $U \approx 8\,\mathrm{m\,s^{-1}}$.
As described above, the basic data sets were time-height cross sections of $w$, $m$, and $\theta$. The following, systematic steps were performed for the data analyses:

- Gridding of all data to vertical resolutions of 15 m and temporal resolutions of 1 s or 10 s, respectively

- Despiking, detrending, and high-pass filtering of the data at each height $z$

- Derivation of the ACFs $A_{w'}(\tau, z)$, $A_{m'}(\tau, z)$, and $A_{\theta'}(\tau, z)$

- Application of the functions defined in Eqs. 8-10 and 11-13 for lags $\geq 1$ to identify the inertial subranges and to investigate over what range of time lags the ACFs should be fitted

- Fit of the ACFs over this range of lags to determine atmospheric variances, the ACF coefficients, and their errors as a function of height





- Characterization of vertical and temporal filter effects and its influence on the shape of the ACFs

- Determination of the profiles of the integral time scales

- Corresponding derivation and evaluation of the power spectra, detection of the inertial subranges for consistency with respect to filter effects and system noise in the frequency domain

- Combination of the profiles of the ACF coefficients and the horizontal wind profile for deriving profiles of TKE dissipation and molecular destruction rates including error propagation

### 275   4.2.1   Gridding, despiking, detrending, and filtering

The time gridding of the data at each height for the chosen averaging time (1 s or 10 s) was performed by block averaging all data points that fell into the corresponding time grid window. Then, the time gridded data were vertically interpolated to 15 m. Please note that this does not change the real vertical resolution of the data, which was 60 m for $w'$ (Träumner et al., 2011) and 100 m for $\theta'$ (Hammann et al., 2015) (block average). For the $m'$ of the WVDIAL, we applied a Savitzky-Golay filter width 280   of 135 m (Späth et al., 2016). We estimated the full-width at half-maximum (FWHM) of an impulse response at the effective filter resolution, which is about 70 % of the filter width or $\approx 95$ m in this case. For the WVDIAL and the TRRL, the vertical resolutions were limited by system noise. At each height, despiking of the hour-long time series was performed by identifying as outliers all data points that fell outside of a $\pm 5$ standard deviation window around the time series median. Only for the $w$ data, we kept the outlier threshold fixed above 1200 m at $\pm 5$ standard deviations of the 1190 m data. Otherwise, because of 285   the large noise of $w$ in the entrainment zone, the outlier threshold would have gotten so large that many noise peaks would not have been filtered out. This procedure was repeated iteratively for the remaining data points until no more outliers were found. The outliers were set to Not a Number (NaN). Due to the high quality of the lidar data analyzed here, despiking became mainly necessary in regions with low SNR, for instance for the WVDIAL and TRRL data in the region of the IL and above. The remaining time series were detrended (subtraction of the time series mean for the $w'$ data and subtraction of a linear fit to 290   the time series for the $m'$ and $\theta'*$ data) and then high-pass filtered using a 30-minute cutoff.

Figure 2 presents the resulting time-height cross section of $w'$ and $m'$ with 1 s resolutions and $\theta'$ with 10 s resolution, respectively. The gap in the $w'$ data around 12 UTC is due to the switch to the conical scan mode for horizontal wind profiling. The data confirm the general challenge of coherent DL vertical wind measurements in the IL and above because the SNR of this type of DL depends strongly on the presence of aerosol particles. Otherwise, the upper panel demonstrates the high resolution 295   and SNR of vertical wind measurements in the CBL. The structure of the turbulent updrafts and downdrafts in the CBL are resolved with great detail. These coherent structures are also clearly visible in the time-height cross sections for $m'$ and $\theta'$, although with higher noise levels. The increase of $\overline{m'}^2$ and $\overline{\theta'}^2$ from the ML to the IL with a maximum in the latter is already visible. $\overline{m'}^2$ is smaller than $\overline{\theta'}^2$ in the ML. The expected high vertical correlation of the data and the correlation between the data can be recognized, particularly for the large updrafts.



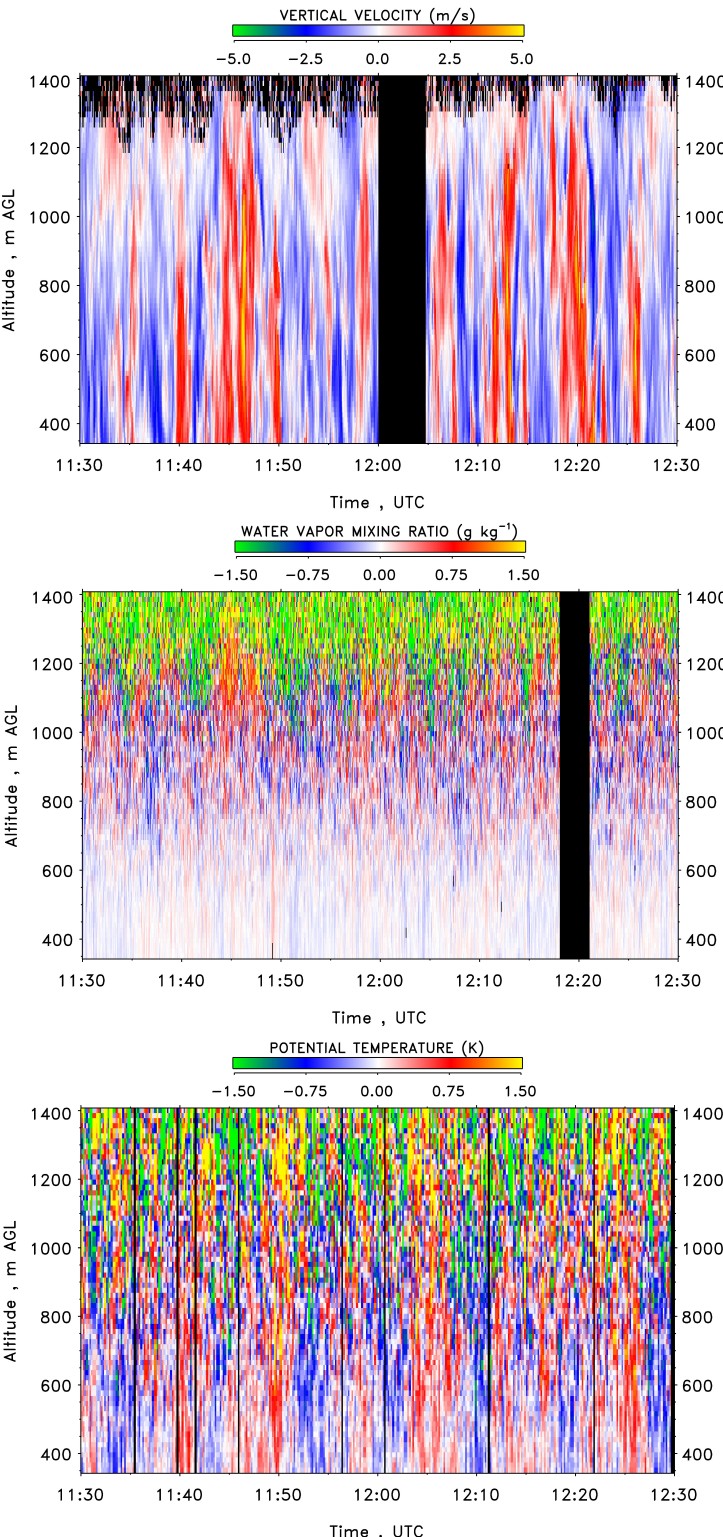

**Figure 2.** Time-height cross section of $w'$ (upper panel, 1 s resolution), $m'$ (middle panel, 1 s resolution), and $\theta'$ (lower panel, 10 s resolution). The black areas indicate measurement gaps or data identified as outliers.





### 4.2.2 Derivation and analyses of the transverse, temporal autocovariance functions

**Vertical wind:**

After the determination of the ACFs of the time series, we studied their shapes for all three variables of interest. We

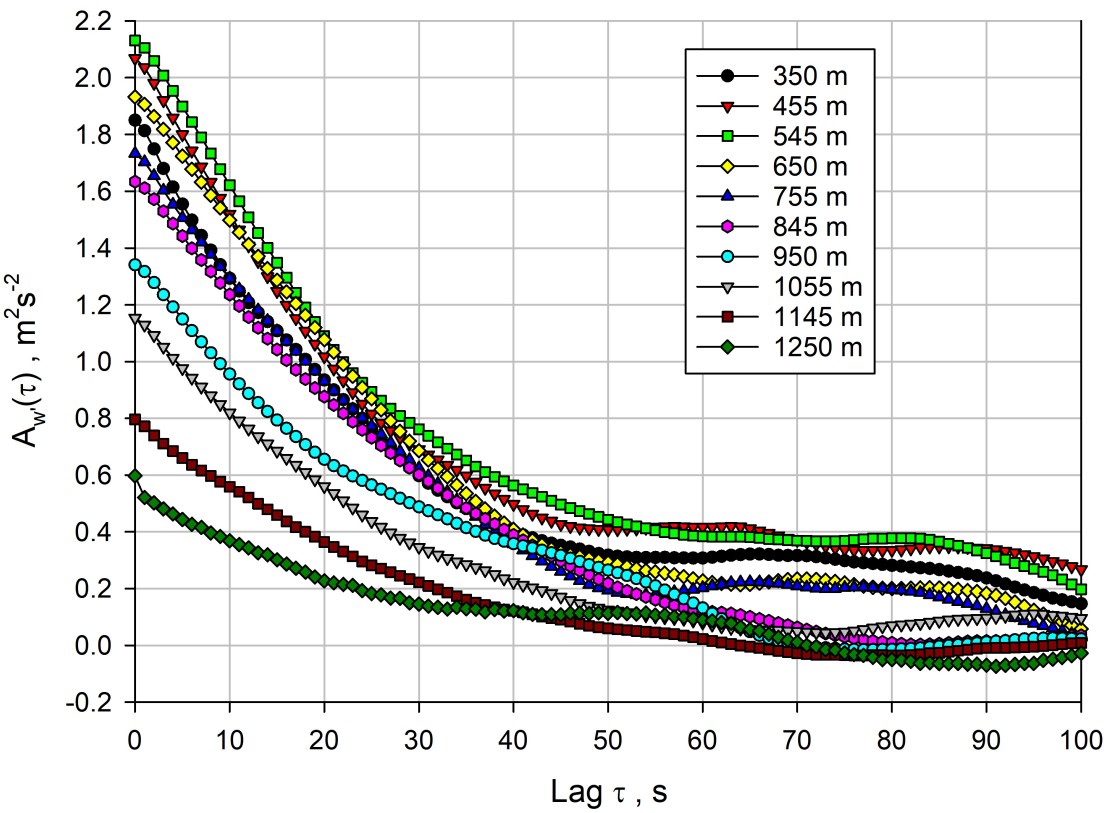

**Figure 3.** $A_{w'}(\tau)$ for a variety of heights using the 1-s data.

started with the examination of $A_{w'}(\tau)$ because the $w'$ data provided the best resolution and SNR in the ML. The results are presented in Fig. 3 between 350 m and 1250 m. The small deviations between lags 0 and 1 confirm the low noise level of the $w'$ measurements. Atmospheric variance associated with mesoscale features affects the ACFs beginning at about lag 25 and beyond. This is also visible in the 3D representation of the ACFs, which is presented in Fig. 4. This figure highlights one of the advantages of the range-resolved lidar measurements because mesoscale influences on the ACFs can be studied as a function of height. It seems that there is more micro- to mesoscale influence at heights between 400-600 m, which may be related to



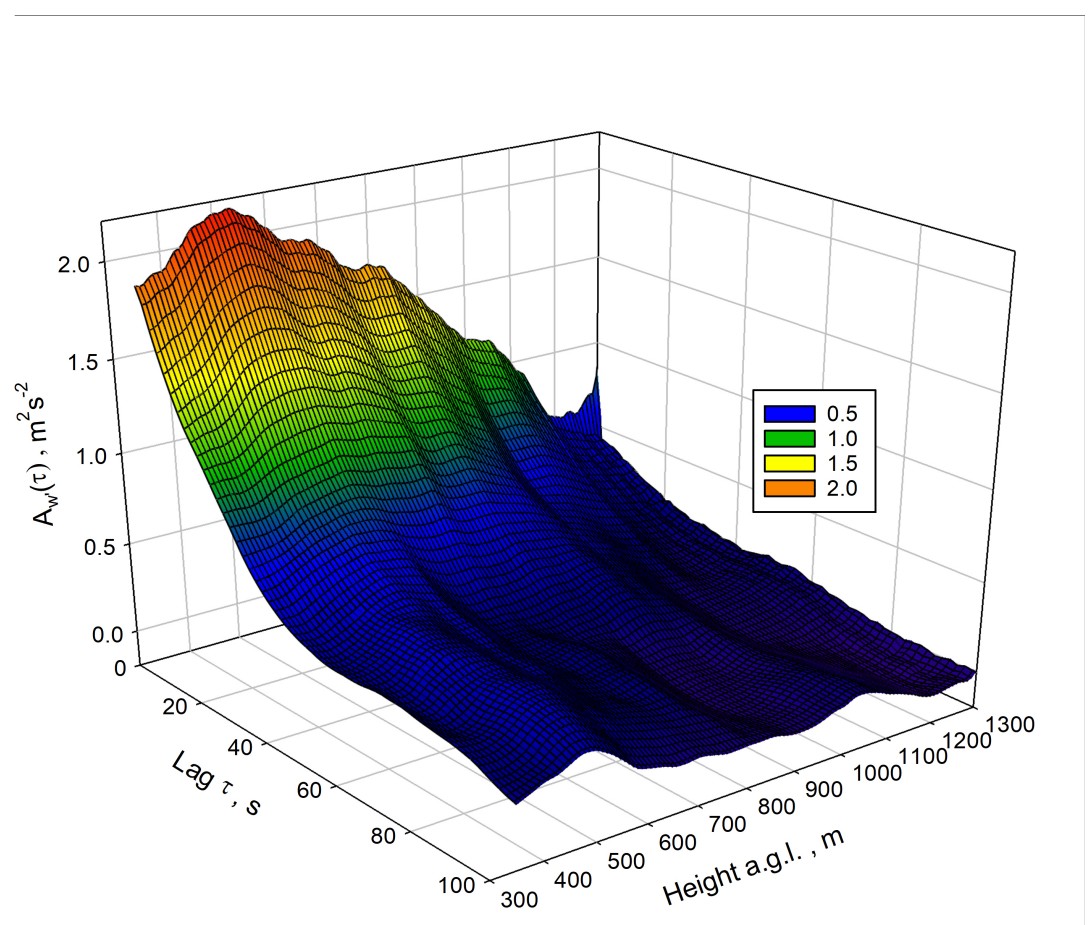

**Figure 4.** 3D plot of $A_{w'}(\tau, z)$ based on the 1-s data.

wind shear in the horizontal wind profile at the same height range. However, as these features do not influence the ACFs at
small lags, a suitable range of lags can be chosen for the fit of the ACF coefficients in the inertial subrange.

The suitable ranges for the fits of the ACFs can be evaluated in more detail using Eqns. 8-10 or 11-13. The latter is visualized
in Fig. 5 for a height of 545 m. For $\overline{w'^2}$ around $2.6\,\mathrm{m^2 s^{-2}}$, the result for $k_w$ is approximately constant for a range between 7 to
25 lags. This result did not change considerably for other heights (not shown) but even extended to lag 30 at 845 m. We assume
that the deviation of the ACF at small lags is due to the spatial filtering of turbulent fluctuations in each range gate of 60 m, as
already studied by Frehlich and Cornman (2002). A minimum fit lag of 7 corresponds well with the approach of Bonin et al.
(2017), who used the along-beam averaging length (60 m in our case) divided by the mean horizontal wind speed ($8$-$9\,\mathrm{m\,s^{-1}}$
here) as a time scale beyond which volume averaging effects can be neglected. Up to about lag 25, no height dependence of
the onset of the mesoscale variability was found so that we identified for this measurement situation an outer time scale of

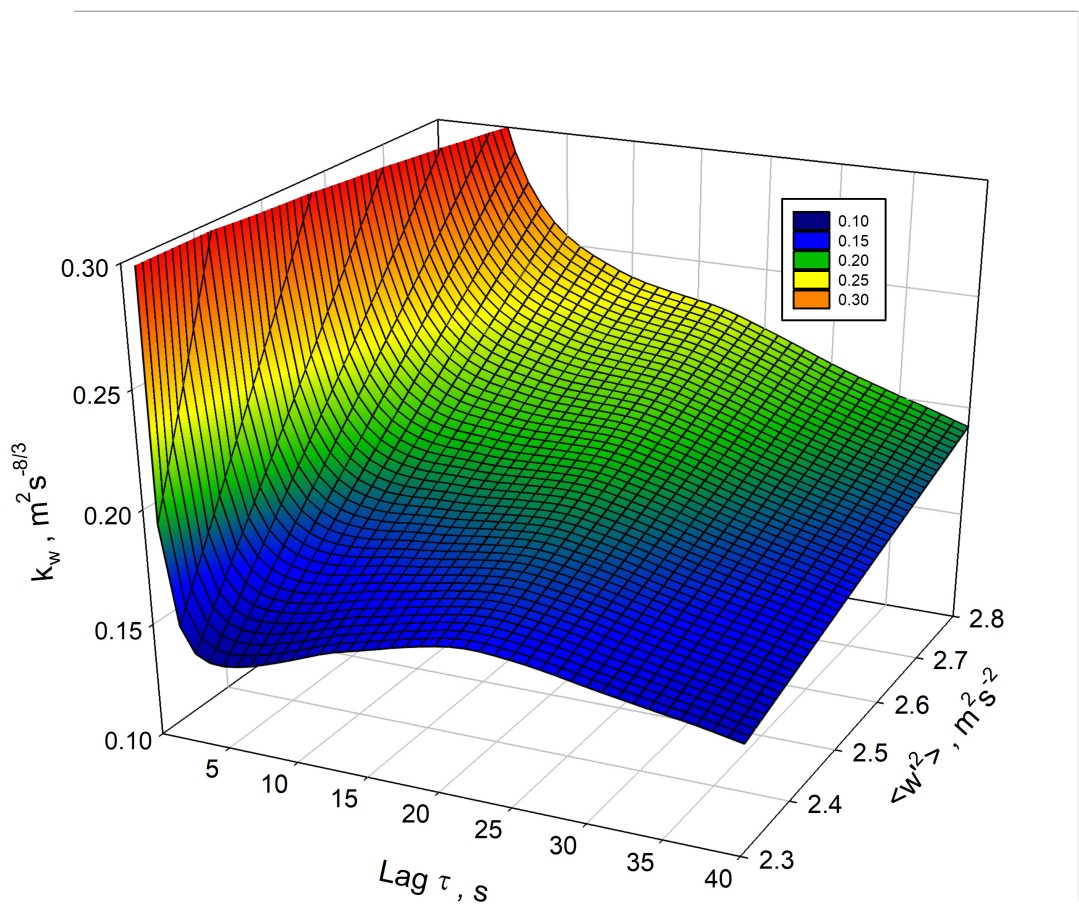

**Figure 5.** Example of the evaluation of the range of suitable lags according to Eq. 11 at a height of 550 m.

turbulence $I_{out} \approx 25$ s limiting the inertial subrange. Therefore, a range of 7 to 25 lags was chosen for the fits of the ACFs for the 1 s data.

Figure 6 shows the corresponding series of power spectra $S_{w'}(\nu)$. There is a nearly height independent spectral maximum around 0.003 Hz and another one at 0.01 Hz. For higher frequencies, the shape of the spectra correspond very well with the inertial subrange up to a frequency of $\nu_I \approx 0.1$ Hz. This behavior is nearly height independent. Beyond this frequency, the roll off of the spectra with a steeper slope is very likely due to the spatial filter effects due to the single shot pulse length. Because of the high SNR of the vertical wind data, it is very useful to perform the vertical wind analyses with a time resolution of 1 s in order to study the ACFs at small lags, which translates to the shape of the power spectra close to the Nyquist frequency of $Ny = 0.5$ Hz. This also allows for estimating the loss of $\overline{w'^2}$ due to the spatial filtering, which is approximately 1 % here. All these effects are consistent with and can also be detected in the temporal domain using the ACFs, as discussed above.



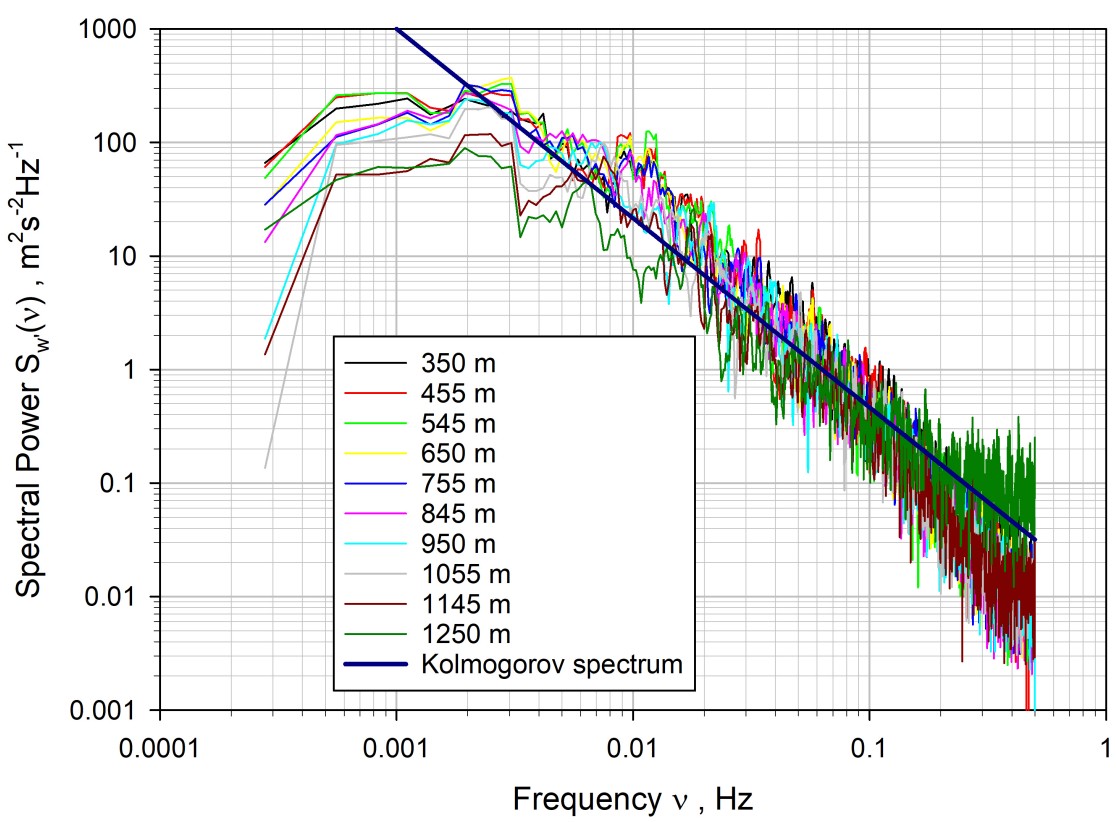

**Figure 6.** Power spectra $S_{w'}(\nu)$ of the vertical wind fluctuations on double-logarithmic scales for a variety of heights. The theoretical -5/3-slope of the inertial subrange is shown as a solid dark blue line.





Therefore, we recommend performing all turbulence analyses in the temporal domain (i.e., using autocorrelations) because the

Fourier transformation does hardly provide additional information and introduces additional noise due to the transformation

process.

It is essential to investigate the sensitivity of the ACFs for both 1 s and for 10 s resolutions in order to find out whether the

coarser time resolutions can be used as well. Figure 7 demonstrates that the ACFs agree very well and that it is possible to use

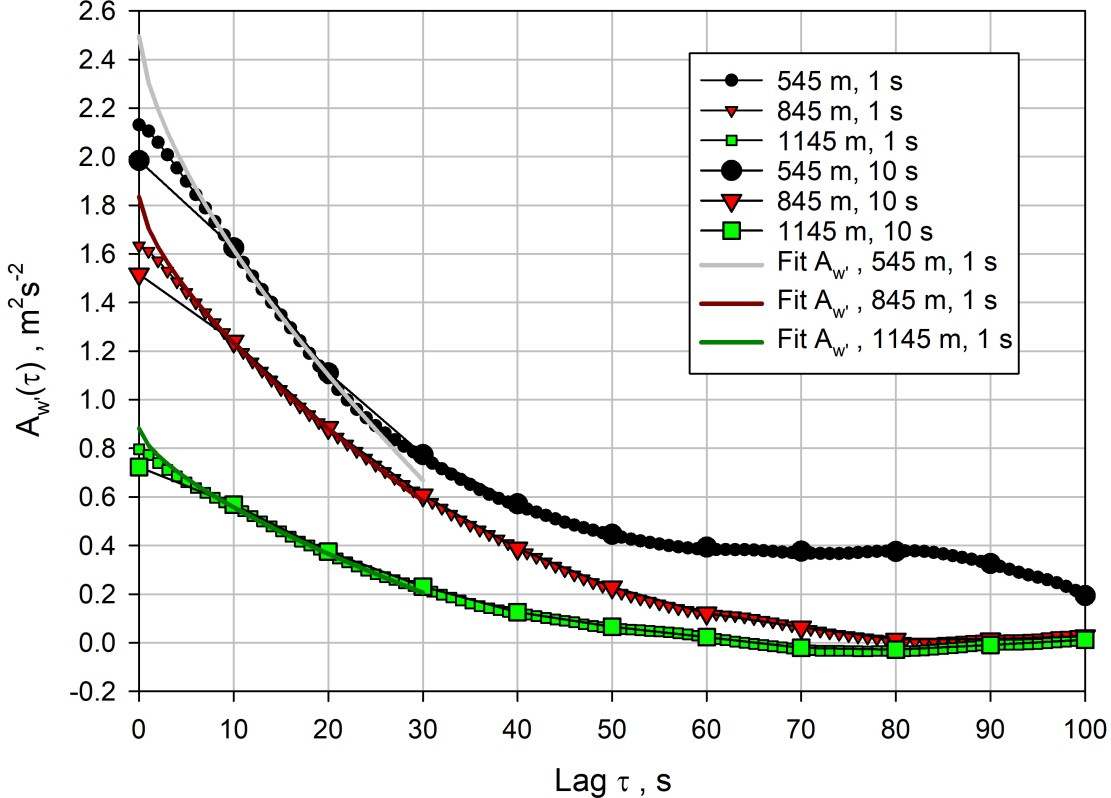

**Figure 7.** Comparison of $A_{w'}$ using 1 s or 10 s resolutions of the vertical wind speed data. Large symbols: 10 s resolution, smaller symbols: 1 s resolution. Colored solid lines: Fits of the theoretical shapes of the ACFs using the 1-s data.

either 1 s or 10 s resolution; however, the number of lags for the fit of the ACFs has to be adapted. Except the situation that a

nonlinear dependency of noise errors is reached, 1-s data analyses are preferred because in this case the resulting noise errors

of the turbulence data are independent of the time resolution but the inertial subrange is better resolved. As mentioned above,

a range of 7-25 lags was used for the 1-s resolution data, corresponding to about 1-3 lags for 10-s data. The total measured

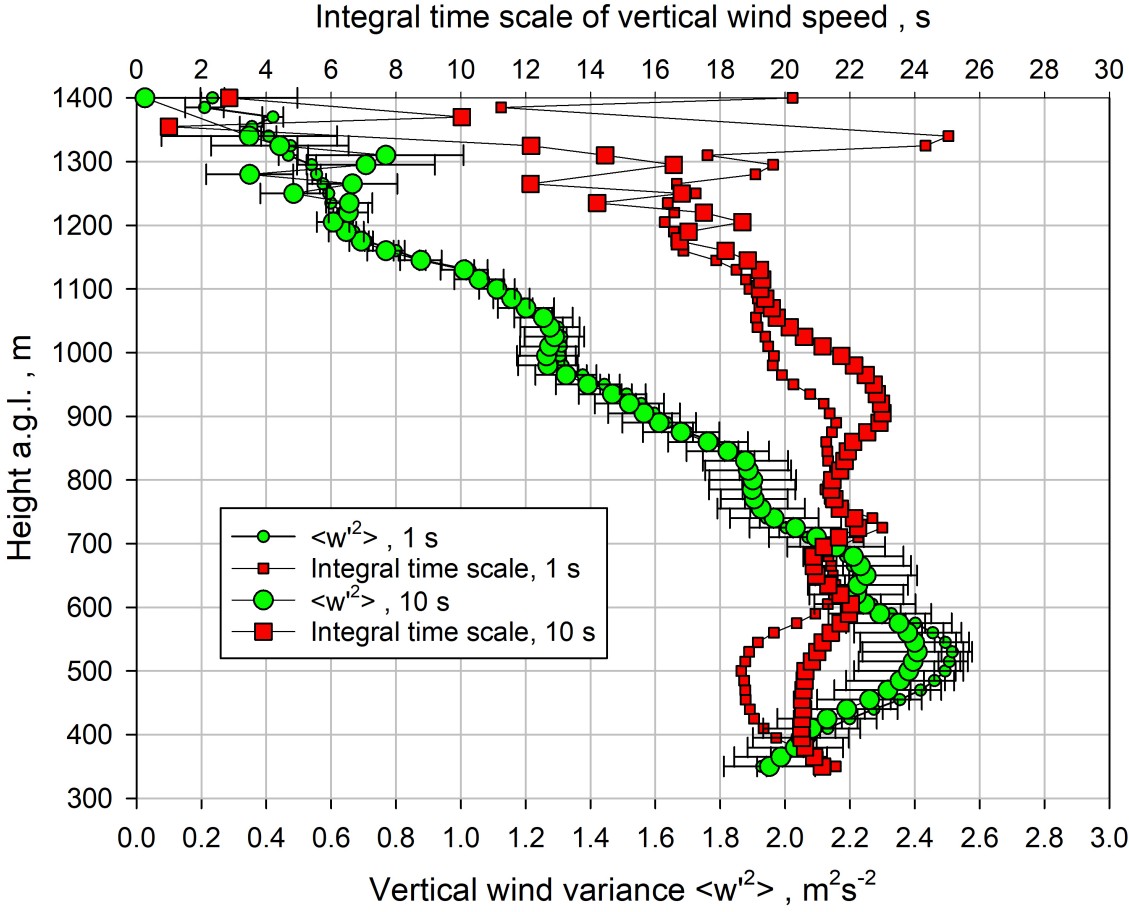

**Figure 8.** Small and large green circles: Profiles of the variance of the vertical velocity fluctuations for 1-s and 10-s data, respectively, including noise error bars. Small and large red squares: The corresponding results for the integral time scales $\mathcal{T}_w$.





variance at lag 0 was smaller for 10-s data because the atmospheric variance at scales between 1 and 10 s is not captured and the noise variance was slightly lower because of the additional time averaging. Figure 7 also shows that the fits of the ACFs

using the chosen lags of 7-25 for the 1-s data (colored solid lines) resulted in a very good agreement with the measured ACFs. For instance, at 845 m, the fit provided an estimated atmospheric variance of $1.62 \, \mathrm{m^2 s^{-1}}$ and $k_w \simeq 0.11 \, \mathrm{m^2 s^{-8/3}}$.

Using these lag ranges for fitting the ACFs, we determined the profiles of the integral time scales $\mathcal{T}_w$ and the variance of the vertical wind fluctuations for both the 1-s and the 10-s data. The results are presented in Fig. 8. The $\mathcal{T}_w$ profiles were derived with Eq. 17, which turned out to be most robust, and agree very well in the ML within a few seconds and amount here to

$\mathcal{T}_w \simeq 20 \, \mathrm{s}$. There seems to be a reduction of $\mathcal{T}_w$ in the IL, which is particularly strong for the 10-s results for $\mathcal{T}_w$. However, it is very likely that this was due to systematic effects by the reduction of data points after quality control. The reduction was less for the 1-s $\mathcal{T}_w$ data where more data points were available in the IL and resulted in an estimate of $\mathcal{T}_w \simeq 16 \, \mathrm{s}$ in this region. These considerations show that for this type of Doppler lidar and its measurement performance, the 1-s data should be used for turbulence analyses.

A similar conclusion can be derived for the profiles of the vertical wind variances $\overline{w'^2}$. The variance profiles agree very well except in the lower ML where the 1-s data derive a slightly higher variance. In the IL, the 10-s data are more noisy, as for $\mathcal{T}_w$, which we attribute again to less data points and resulting poorer sampling statistics. In the IL, we derived a decay of the vertical wind variance resulting in values of $\overline{w'^2} \simeq 0.4 - 0.6 \, \mathrm{m^2 s^{-2}}$. The maximum of the variance profile is located at $530 \, \mathrm{m}$, which corresponds to $0.4 \, z/z_i$. The variance maximum is approximately $2.5 \, \mathrm{m^2 s^{-2}}$. This is in good agreement with previous results

(Lenschow et al., 2000; Lothon et al., 2006, 2009). These results confirm that the 1-s data should be preferred for vertical wind turbulence analyses so that we proceeded with these data for the derivation of turbulence profiles.

**Water vapor mixing ratio:**

For the mixing ratio fluctuations, we are challenged by the trade-off between the accuracy and resolution of the WVDIAL measurements. The non-linear reduction of system noise as a function of the spatial resolutions is explained in Wulfmeyer

et al. (2015), Eq. 28, whereas the reduction of noise due to temporal averaging behaves the same as for the DL for high SNR and becomes also non-linear at low SNR, for example due to background signal subtraction. Though to our knowledge the UHOH WVDIAL currently the active remote sensing system to measure water-vapor profiles and their fluctuations with highest resolution and accuracy (Späth et al., 2016; Behrendt et al., 2020), it is still necessary to find the best trade-off with respect to spatial and temporal resolutions to measure the atmospheric fluctuation in the mixing ratio. Therefore, we investigated

the measurements using 1 s and 10 s resolutions. However, we kept a vertical resolution of $100 \, \mathrm{m}$ (see above section 4.2.1). At higher vertical resolution, the data became too noisy and we wanted to avoid stronger spatial filtering effects at coarser resolution.

The ACFs are presented in Fig. 9. This figure confirms that at 1-s resolution, the measurements in the near-range are not strongly influenced by noise but the variance in the WVMR is also very small. In the ML, the mixing ratio variance increases

and reaches a maximum in the IL (in contrast to the vertical wind variance) but this is also true for the system noise. Fortunately, the combination of 1-s or 10-s temporal and 70-m spatial resolutions provide a compromise so that the atmospheric variance can still be recovered throughout the CBL. This is reflected in Fig. 9. Though the total variances of the time series and the

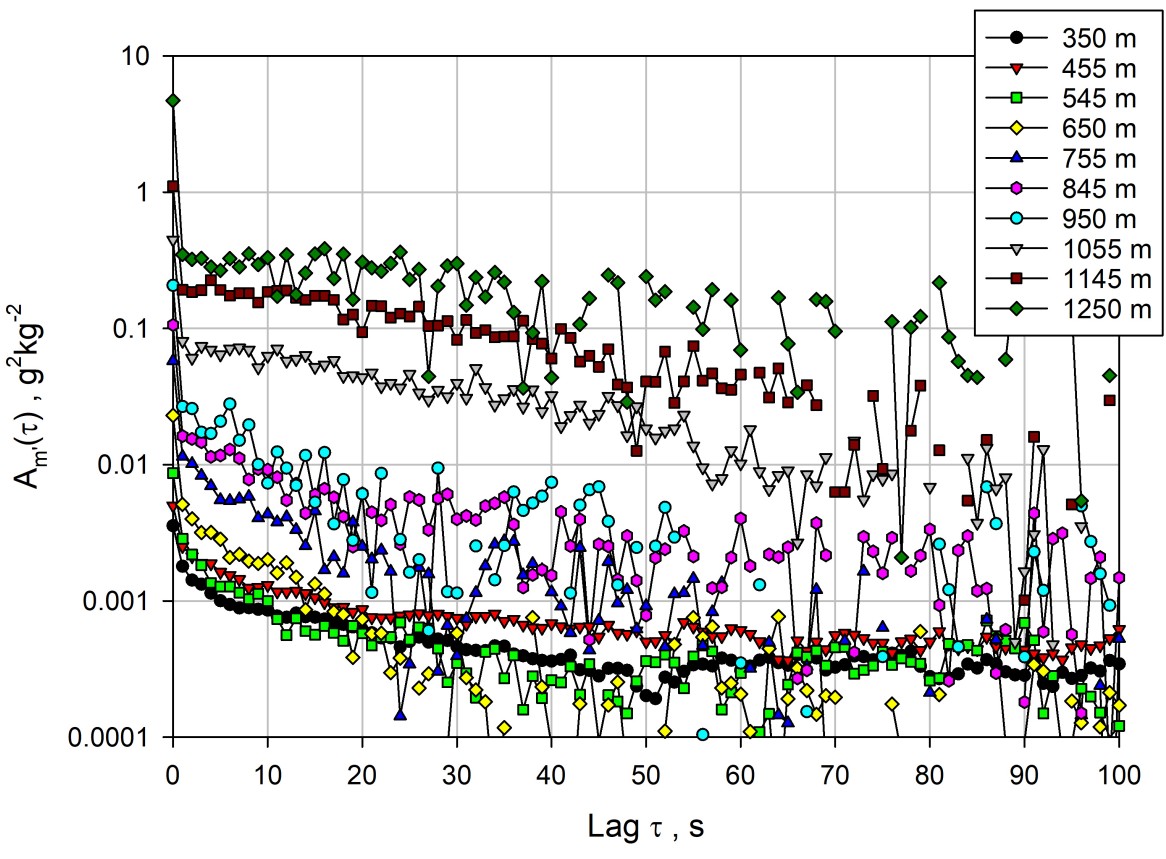

**Figure 9.** $A_{m'}(\tau)$ for a variety of heights. Please note that the ACFs are shown on a logarithmic scale in order to capture the large dynamic range of the WVMR variances in dependence of height.



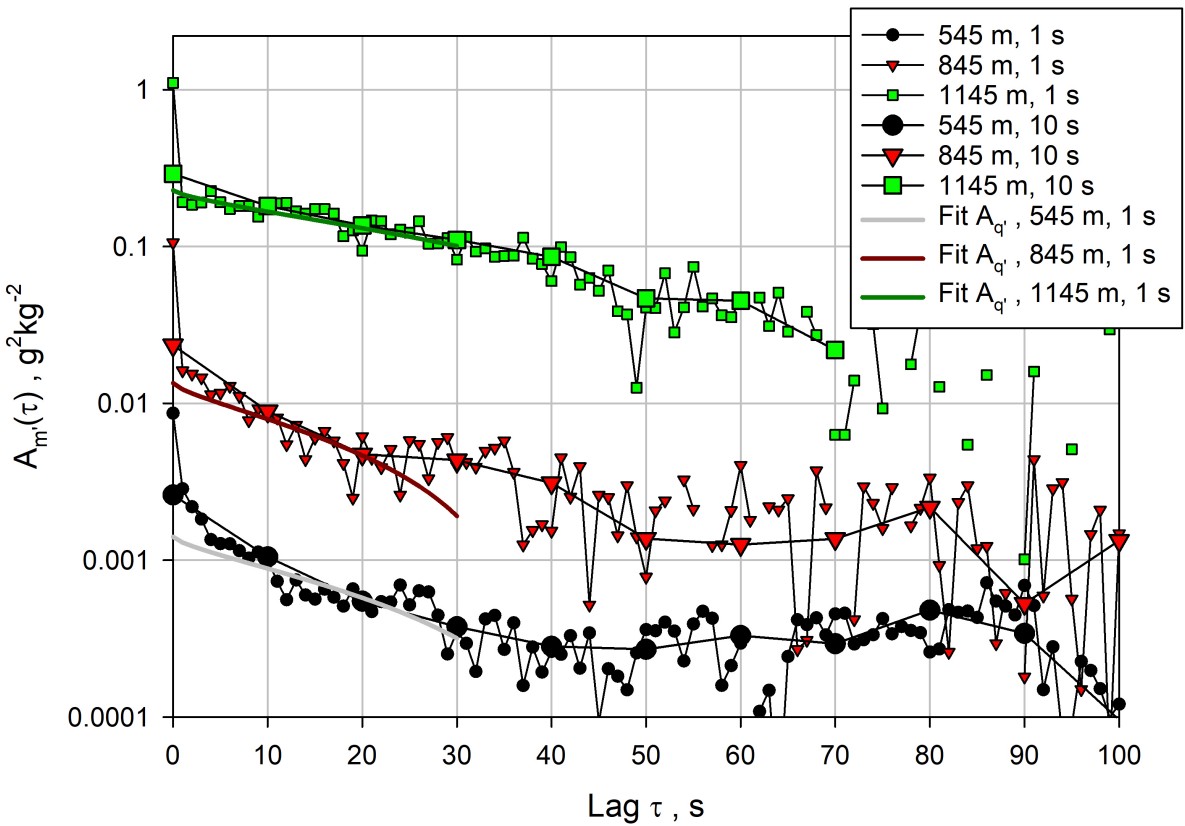

**Figure 10.** Comparison of $A_{m'}(\tau)$ for three heights using either 1-s or 10-s data. Solid lines: Fits of the theoretical shapes of the ACFs.

contribution of the noise variances are visible the marked increase at lag 0, is still possible to separate noise and atmospheric variance by the extrapolation of the ACFs to lag 0.

The comparison between the ACFs computed using the 1-s and 10-s resolutions is presented in Fig. 10. As for the vertical wind data, a range of 7-25 lags for the 1-s data and 1-3 lags for the 10-s data turned out to be optimal for the fit of the ACFs. The fits using the 1-s data are also shown on this plot up to lag 30 and agree well with the theoretical shapes of the ACFs. The power spectra for the 1-s data are shown in Fig. 11. There is an apparent deviation from the expected slope in the inertial subrange at high frequencies. However, these deviations are almost entirely due to the high noise levels, which led to a deviation from the

-5/3-slope on the logarithmic scale. This is substantiated by the fits (solid green, dark red, and grey lines) of the spectra to the Kolomogorov function $\nu^{-5/3}$ (solid blue line) in the inertial subrange plus the noise floor. These fits show that the WVDIAL still resolves atmospheric fluctuations over the 0.005-0.05 Hz range but these are masked by the larger noise floor. Therefore,



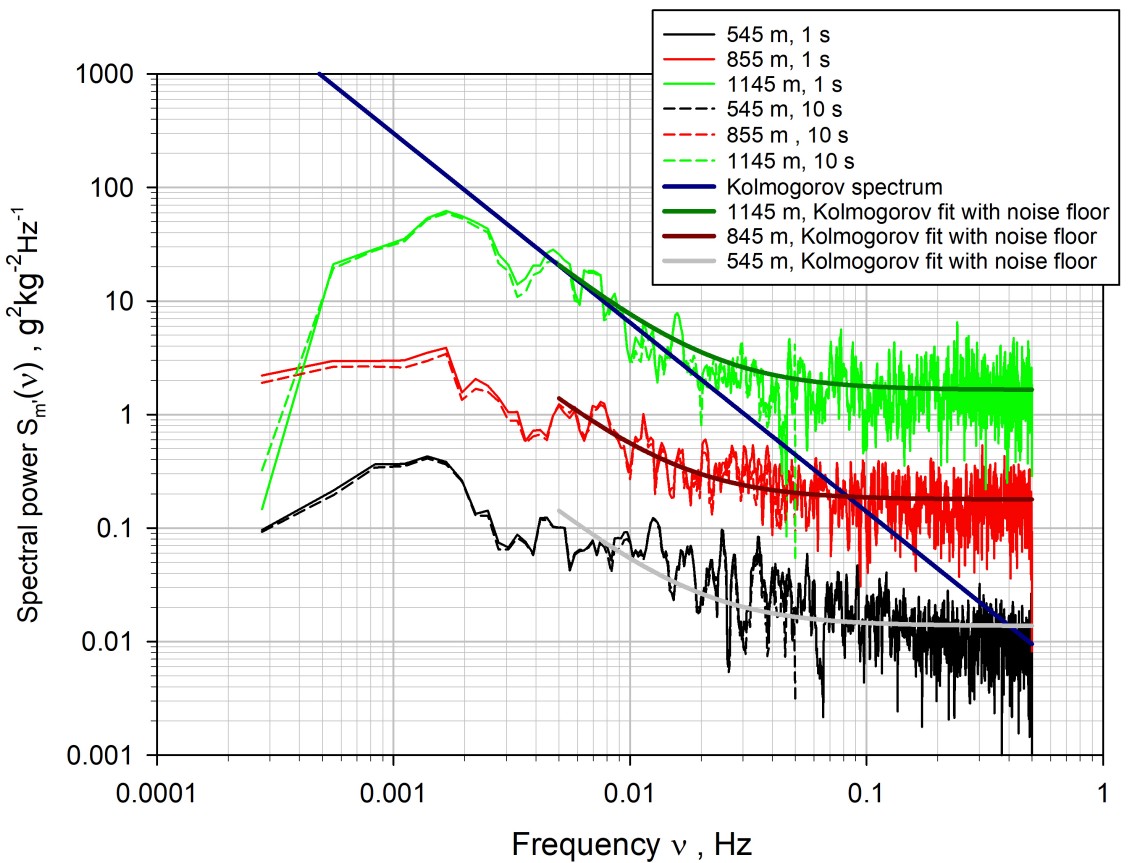

**Figure 11.** Power spectra of the WVMR fluctuations using 1-s data. The 10-s spectra are very consistent (not shown). The fits of the Kolmogorov spectrum with the noise floor are also shown.



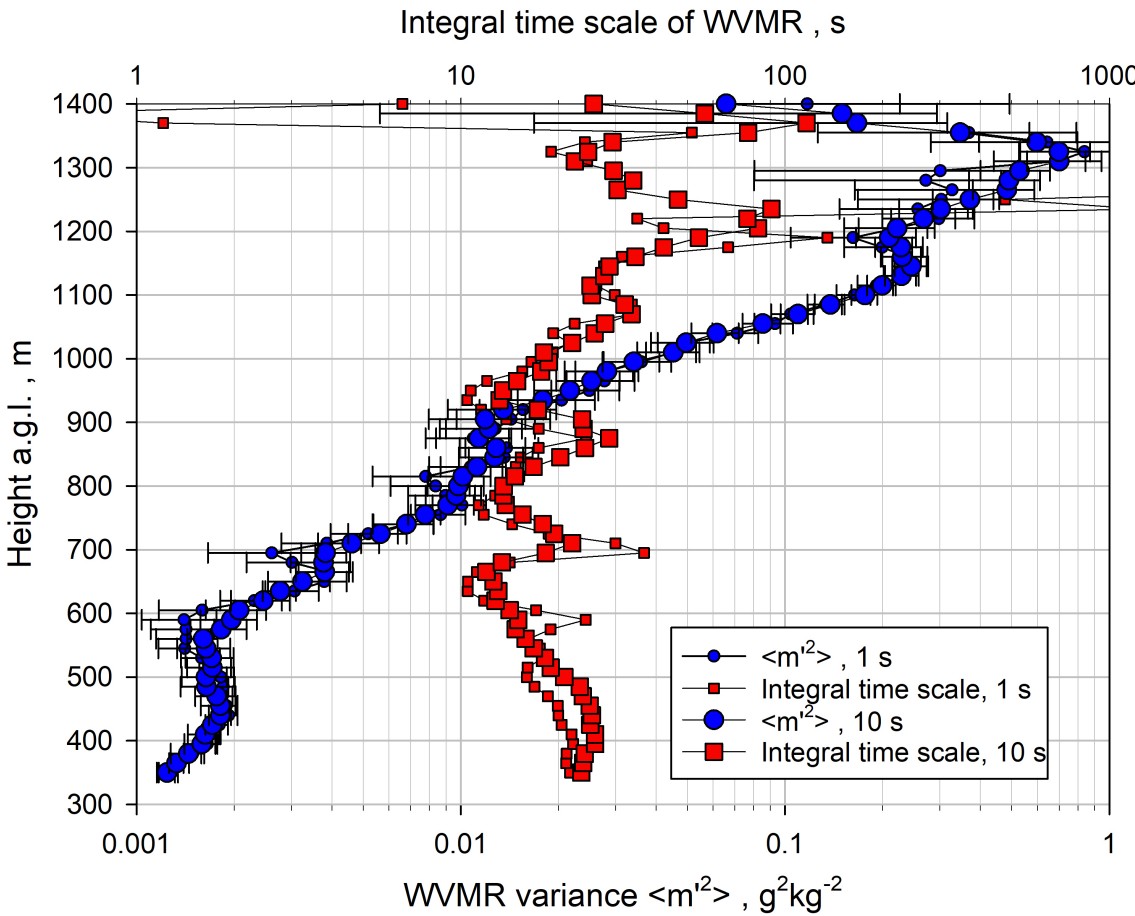

**Figure 12.** Small and large blue circles: Profiles of the variance of the WVMR fluctuations for 1-s and 10-s data, respectively. Small and large red squares: The corresponding results for the integral time scales $\mathcal{T}_m$. The vertical variability of $\mathcal{T}_m$ is likely due to the division by small and noise values of $k_m$ (see Eq. 17).





in all cases, the inertial subrange is resolved but less obvious due to the higher noise level. Particularly, this is important for the critical measurements in the IL (i.e., the data in green at 1145 m) where the molecular destruction rate reaches a maximum,

which at least should be quantified by this measurement method.

Using these lag ranges for fitting the ACFs, we determined the profiles of the integral time scales $\mathcal{T}_m$ and the variance of the WVMR fluctuations $\overline{m'^2}$ for both the 1-s and the 10-s data. The results are presented in Fig. 12. Both data sets are plotted on a log scale to visualize the strong non-linear increase of $\overline{m'^2}$ in the ML and the capability of the WVDIAL to recover this large dynamic range of atmospheric variances. In the ML, the determination of $\mathcal{T}_m$ is strongly influenced by system noise at

low variance values because in this case also $k_m$ is small resulting in a large uncertainty of $\mathcal{T}_m$ (see Eq. 17). Nevertheless, the 1-s and 10-s are consistent within a few seconds. In average in the ML, $\mathcal{T}_m \simeq 20\,\mathrm{s}$, similar to the $\mathcal{T}_w$ of the vertical wind. However, in the IL, the 10-s $\mathcal{T}_m$ increases significantly to around $50\,\mathrm{s}$ whereas the 1-s $\mathcal{T}_m$ show strong fluctuations also to very small values. Here, we consider the 10-s data as more realistic because the noise level of the 1-s data increases in a non-linear manner above the CBL so that it is better to use longer time averages.

The profiles of $\overline{m'^2}$ using 1-s or 10-s data are very consistent in the entire CBL confirming our methodological approach to determine $\overline{m'^2}$ using different time resolutions. Using the extrapolation of the ACF to lag 0, reasonable estimates of $\overline{m'^2}$ are derived even if the inertial subranges are not fully resolved. Also, over the entire CBL, there is almost no evidence of loss of variance due to temporal filtering effects. With the WVDIAL, it is possible to derive $\overline{m'^2}$ and its non-linear increase over nearly three orders of magnitude in the ML. In the IL, $\overline{m'^2}$ reached a maximum very close to $z_i$. Interestingly, we found two maxima

in the $\overline{m'^2}$ profile, one at 1150 with a value of $\overline{m'^2} \simeq 0.25\,\mathrm{g^2 kg^{-2}}$ and the other one at $z_i$ with a value of $\overline{m'^2} \simeq 0.75\,\mathrm{g^2 kg^{-2}}$. In the following, we used the the 1-s and 10-s $k_w$ profiles in combination with the 10-s $k_m$ profiles further turbulence profiles because the latter provided more robust results in the IL. **Potential temperature:** For the potential temperature fluctuations, it is particularly difficult to derive structures in the ACF and the power spectra due to the large noise level in the TRRL observations. Though the improvement of TRRL towards the resolution of temperature fluctuations was substantial in recent

years (see, e.g., Lange et al. (2019)), it is still only possible to study the data with 10-s resolution. Nevertheless, as the temporal resolution of 10 s was still sufficient to use lags 1-3 for the extrapolation using the WVDIAL data and its results did not show significant differences in the determination of variances and temporal integral scales between the 1-s and 10-s data, we applied this technique also to the 10-s TRRL data for providing realistic profiles of $\mathcal{T}_\theta$, $\overline{\theta'^2}$, and $k_\theta$.

The corresponding ACFs are presented in Fig. 13 and the fits to the theoretical shapes of the ACFs for three heights in

Fig. 14. Over all ranges, the derivation of the atmospheric temperature variance and the slope of the ACFs are at the detection limits. This is also visible in the power spectrum in Fig. 15. A range that corresponds to the inertial subrange is barely visible. However, this is also not necessary, as long as the 10-s data contain the major contributions of $\overline{\theta'^2}$, as shown for the WVDIAL data. Using lags 1-3 for fitting the ACFs, we determined the profiles of the integral time scales $\mathcal{T}_\theta$ and the variance of the potential temperature fluctuations. The results are presented in Fig. 16. For the same reasons as for $\mathcal{T}_m$, $\mathcal{T}_\theta$ is strongly affected

by system noise at low variance values. In spite of the large fluctuations, $\mathcal{T}_\theta$ seems to be larger than $\mathcal{T}_w$ and $\mathcal{T}_m$ in the lower ML and decreases towards the IL. Our estimate of $\mathcal{T}_\theta$ ranges between $\simeq 30$,s in the ML and $\simeq 14$ in the IL. In contrast to $\mathcal{T}_m$,



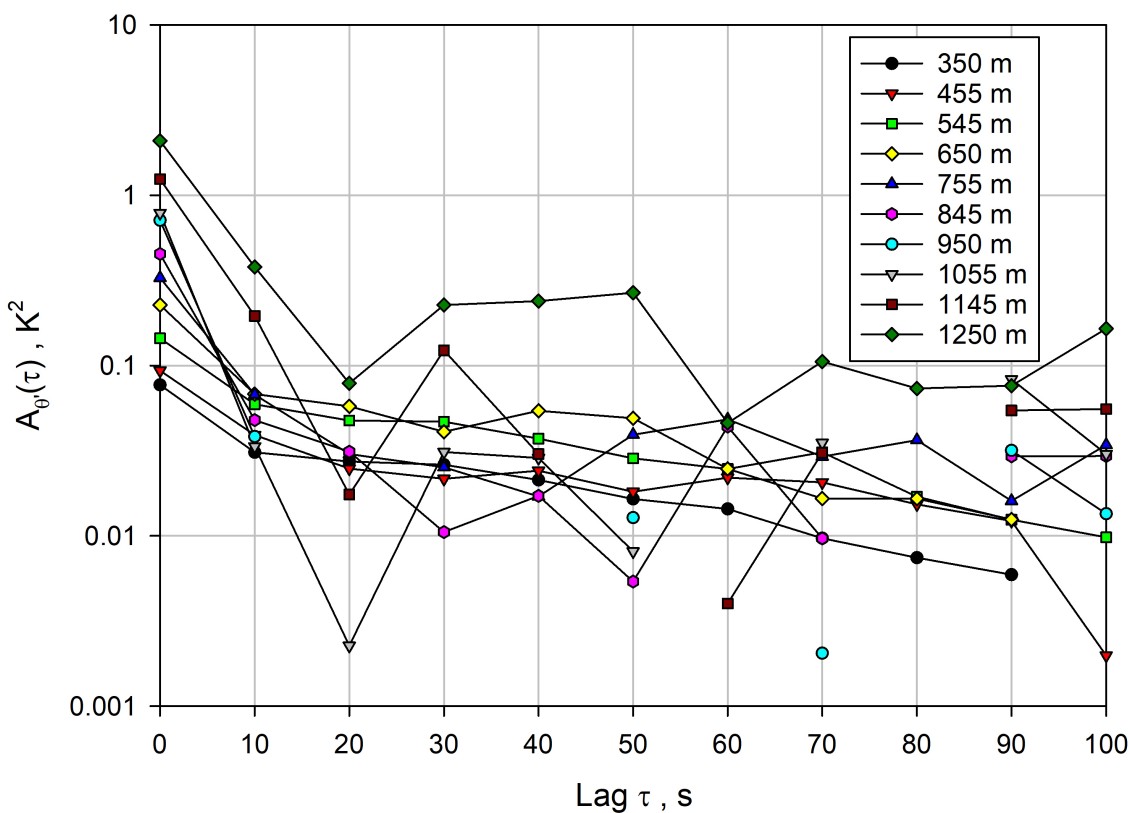

**Figure 13.** $A_{\theta'}(\tau)$ for a variety of heights using a resolution of 10 s.

there seems to be a reduction of $\mathcal{T}_\theta$ from ML towards the IL. In the major part of the ML, all $\mathcal{T}$ s for $w'$, $m'$, and $\theta'$ are similar for the 1-s and 10-s data and range between $\simeq 20 - 30$ s.

Using the fits of the ACFs, we are able to detect the expected small $\overline{\theta'^2}$ in the ML and its increase to a maximum in the IL reaching a statistically significant $\overline{\theta'^2} \simeq 1.1 \, \text{K}^2$ very close to $z_i$. In contrast to $\overline{m'^2}$, we did not find a strong non-linear increase of $\overline{\theta'^2}$ in the ML but it remained around $\simeq 0.1 \, \text{K}^2$ over almost the entire part of the ML. As for the $\overline{m'^2}$ profile, there seems to be another slight peak of the variance at 1020 m, which corresponds to approximately $0.8 \, z/z_i$, with a value of $\overline{\theta'^2} \simeq 0.2 \, \text{K}^2$.

A comparison of the integral length scales is presented in Fig. 17. There are strongly deviations in the lower ML and the IL; however, in the center of the ML, the integral length scales are similar and amount $\simeq 160$ m. As for $m'$, we applied the the 1-s and 10-s $k_w$ profiles in combination with the 10-s $k_\theta$ profiles further turbulence profiles because only the latter provided robust results in the CBL.



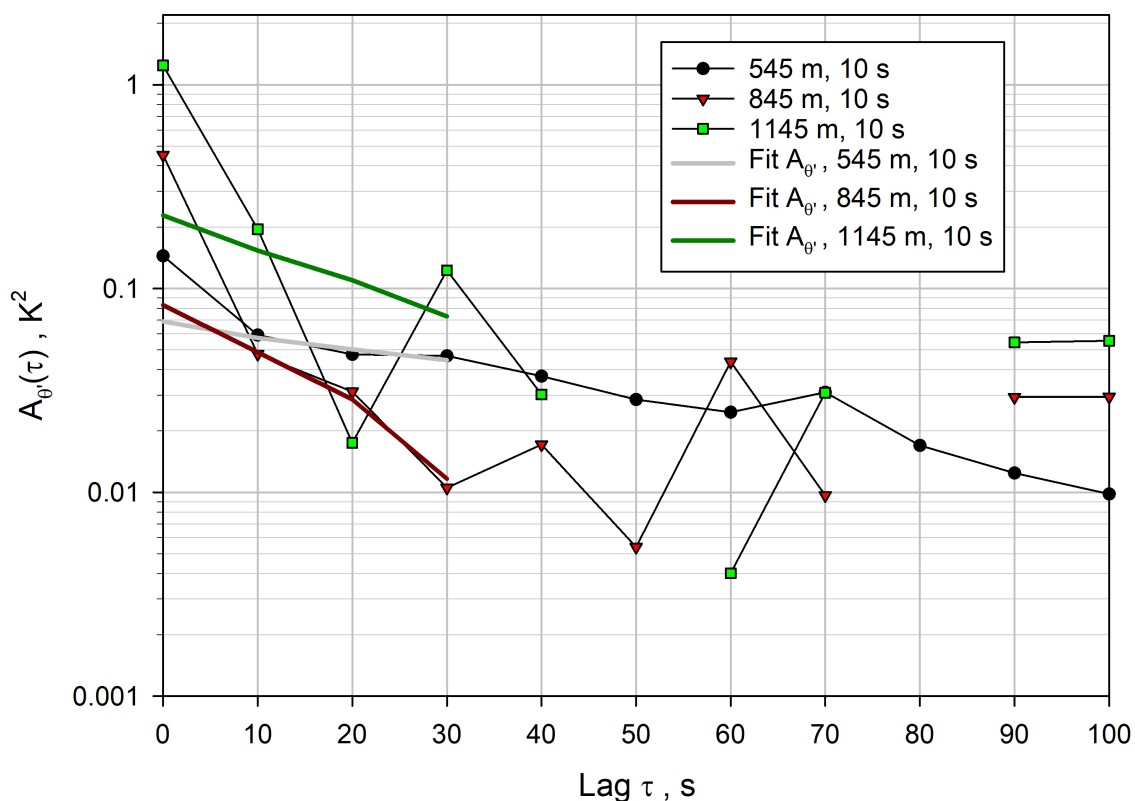

**Figure 14.** $A_{\theta'}(\tau)$ for 10 s at three heights with the corresponding fits of the theoretical shape of the ACFs.

## 5   Profiles of TKE dissipation and molecular destruction rates

### 5.1   TKE dissipation

The derivation of the profiles of the ACF coefficient $k_w$ and the TKE dissipation $\epsilon$ (see Eq. 18) using the 1-s and 10-s data

are presented in Fig. 18. Similar to $\overline{w'^2}$, both $k_w$ and $\epsilon$ reach a maximum near 530 m, which corresponds to approximately

0.4 $z/z_i$. Up to this height, likely due to filter and noise effects, the difference between the 1-s and 10-s data is largest, whereas

the results are very similar in the rest of the ML. At the maximum, $\epsilon \simeq 1 \cdot 10^{-2}\,\mathrm{m^2 s^{-3}}$ for the 1-s data and $\epsilon \simeq 8.5 \cdot 10^{-3}\,\mathrm{m^2 s^{-3}}$

for the 10-s data, respectively. At this height, $k_w \simeq 0.19\,\mathrm{m^2 s^{-8/3}}$ for the 1-s data and $k_w \simeq 0.17\,\mathrm{m^2 s^{-8/3}}$ for the 10-s data,

respectively. Both $\epsilon$ and $k_w$ decrease towards the IL to values around $\epsilon \simeq 1 \cdot 10^{-3}\,\mathrm{m^2 s^{-3}}$ and $k_w \simeq 0.04\,\mathrm{m^2 s^{-8/3}}$ at 1300 m,

respectively. In the IL, as expected from the analyses of the variances and the integral time scales, the 10-s data showed a high



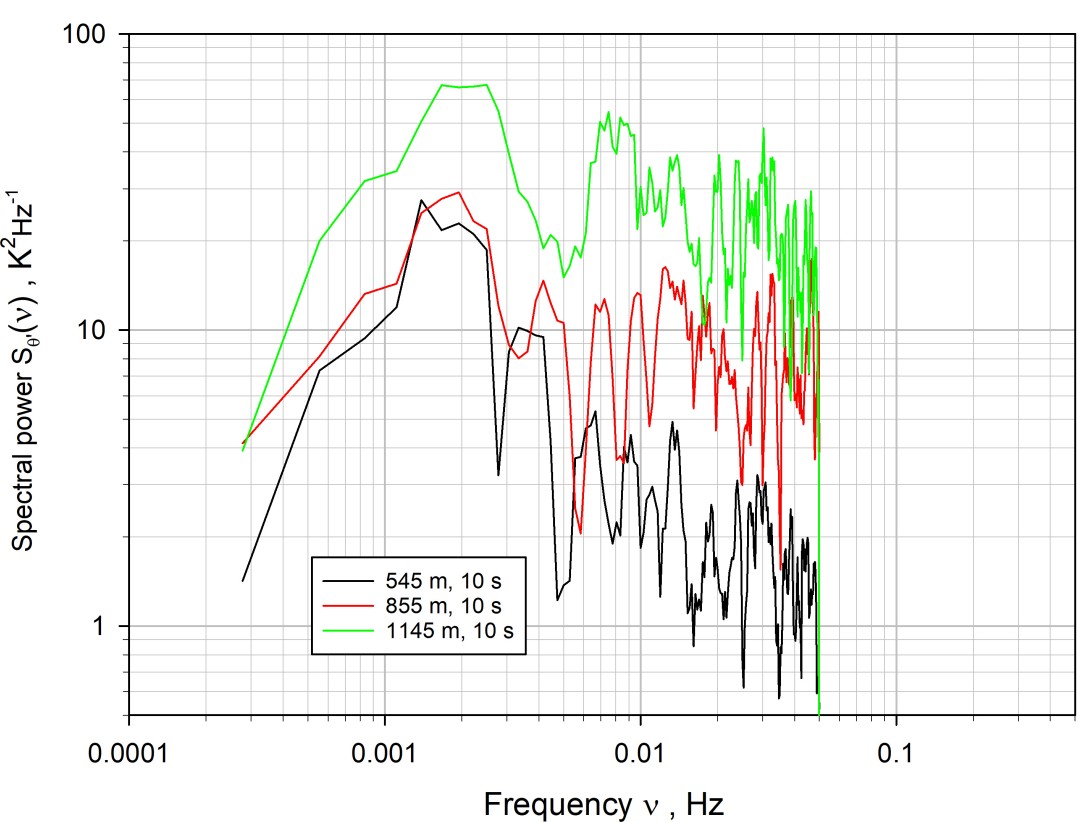

**Figure 15.** Power spectra $S_{\theta'}(\nu)$ of the potential temperature fluctuations on double-logarithmic scales for a variety of heights. The theoretical -5/3-slope of the spectra is hardly visible due to the high noise level of the data and the need to use 10-s data. However, a large part of the atmospheric variances are still recovered over this spectral range.

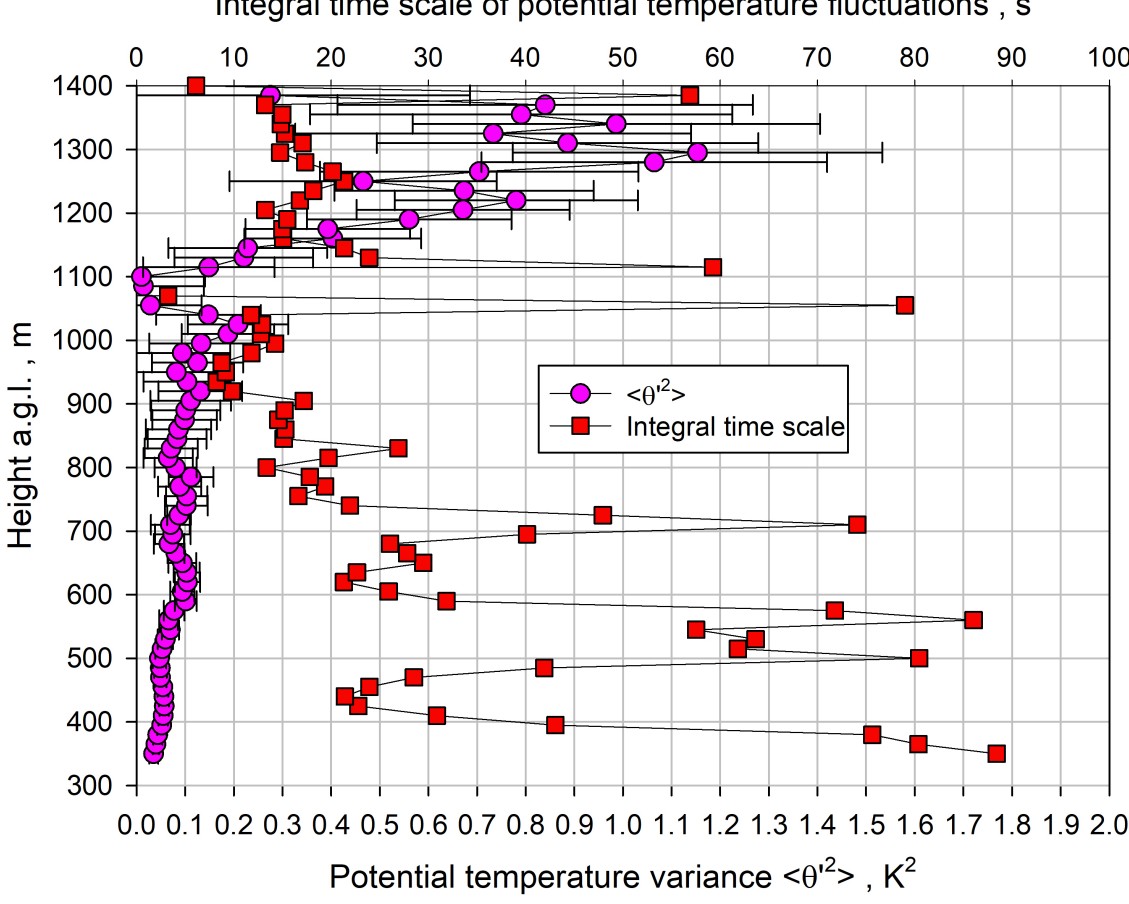

**Figure 16.** Pink circles: Profiles of the variance of the potential temperature fluctuations for 10-s data. Red squares: The corresponding results for the integral time scale.

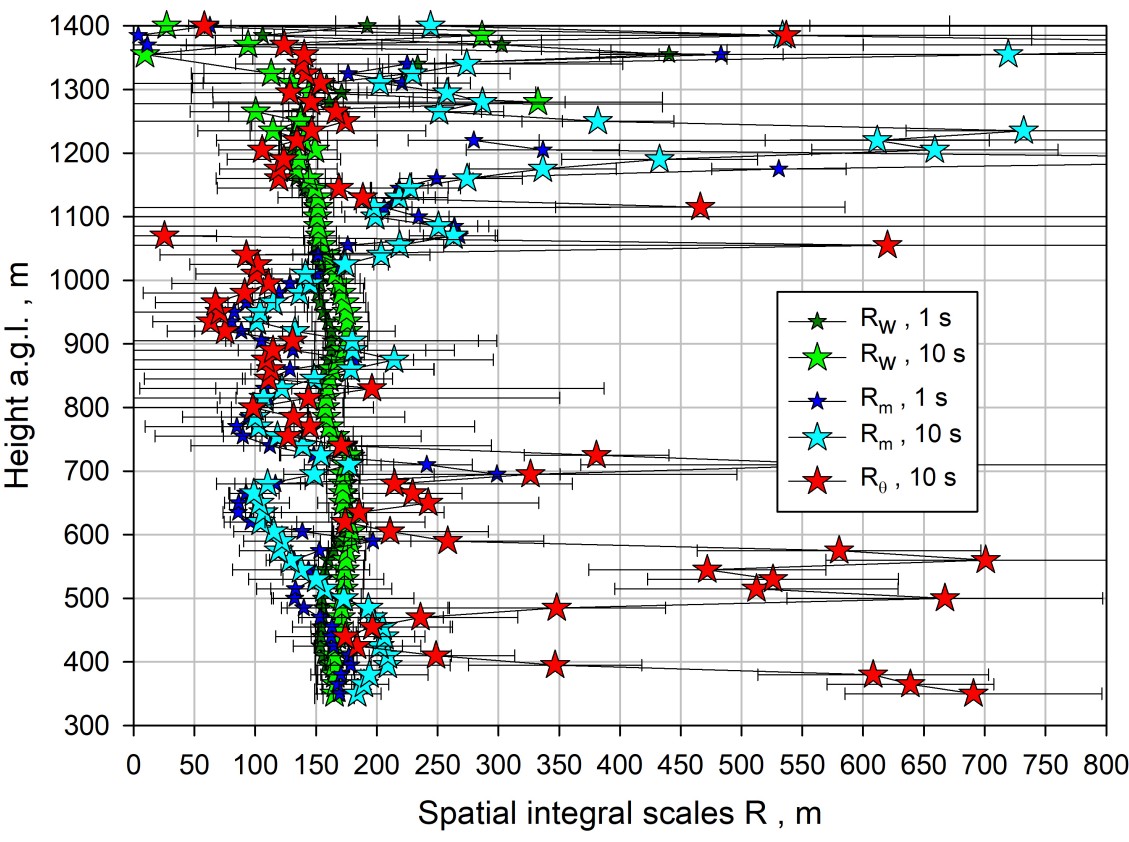

**Figure 17.** Comparison of the integral length scales $\mathcal{R}_w$, $\mathcal{R}_m$, and $\mathcal{R}_\theta$ including error bars due to noise error propagation.

variability whereas the 1-s data still permitted robust derivations of $\epsilon$ and $k_w$. As also in the ML a more accurate determination of turbulence profiles can be expected due to the better resolution of the inertial subrange, we used the 1-s profiles for the derivations of molecular destruction rates below.

For parameterizations of $\epsilon$, it can be related to the vertical wind variance profile $\overline{w'^2}$. The results are presented in Fig. 19 with the theoretical function from Eq. 19. For this fit we used all the data and inserted a mean integral length scale averaged over all data and the entire CBL, which resulted in $\mathcal{R}_w \simeq 162.5\,\mathrm{m}$ (see Fig. 17). Please note that the averages of $\mathcal{R}_w$ was quite consistent between the 1-s ($\simeq 160\,\mathrm{m}$) and 10-s data ($\simeq 165\,\mathrm{m}$) and decreased only by 20 s towards the IL so that it was valid to use just an average value for $\mathcal{R}_w$. Thus, we achieved

$$\epsilon \simeq \frac{2}{5}\frac{1}{160\,\mathrm{m}}\left(\overline{w'^2}\right)^{3/2} \quad , \tag{25}$$

which turned out to be in excellent agreement with our measurements.



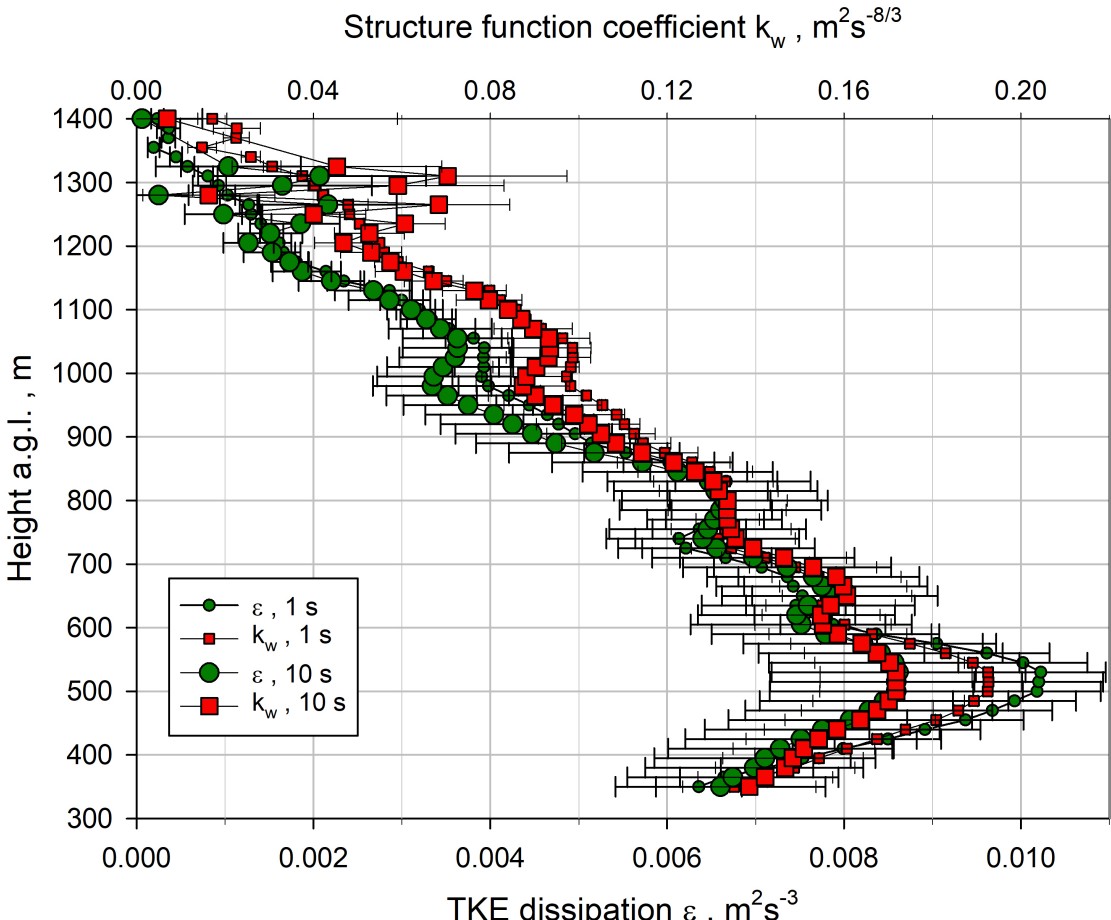

**Figure 18.** Small and large dark green circles: Profiles of TKE dissipation $\epsilon$ derived with 1-s and 10-s resolutions, respectively. Small and large red squares: Corresponding profiles of the ACF coefficients $k_w$ with 1-s and 10-s resolutions, respectively.



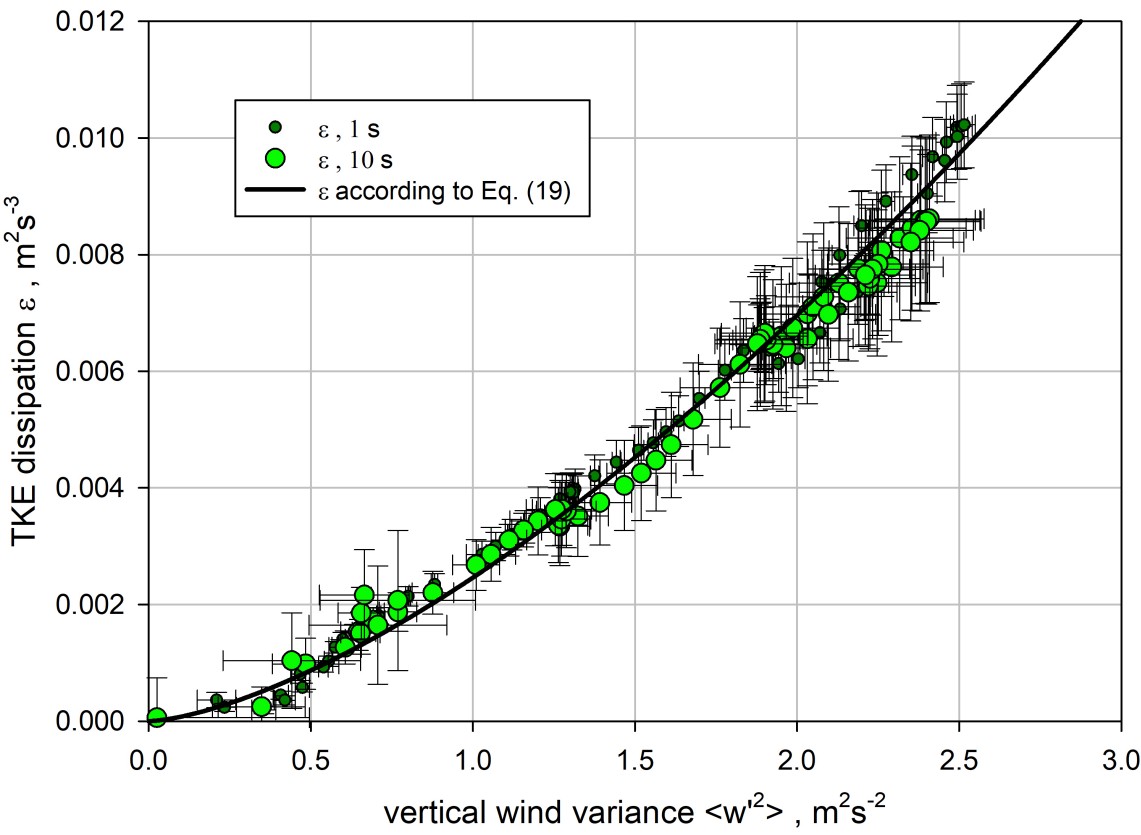

**Figure 19.** Small dark green circles: TKE dissipation $\epsilon$ derived with 1 s resolution. Large light green circles: $\epsilon$ derived with 10 s resolution. Solid black line: Fit using Eq. 19 combining all data and using the a mean spatial integral length scale $\mathcal{R}_w \simeq 162.5$ m.

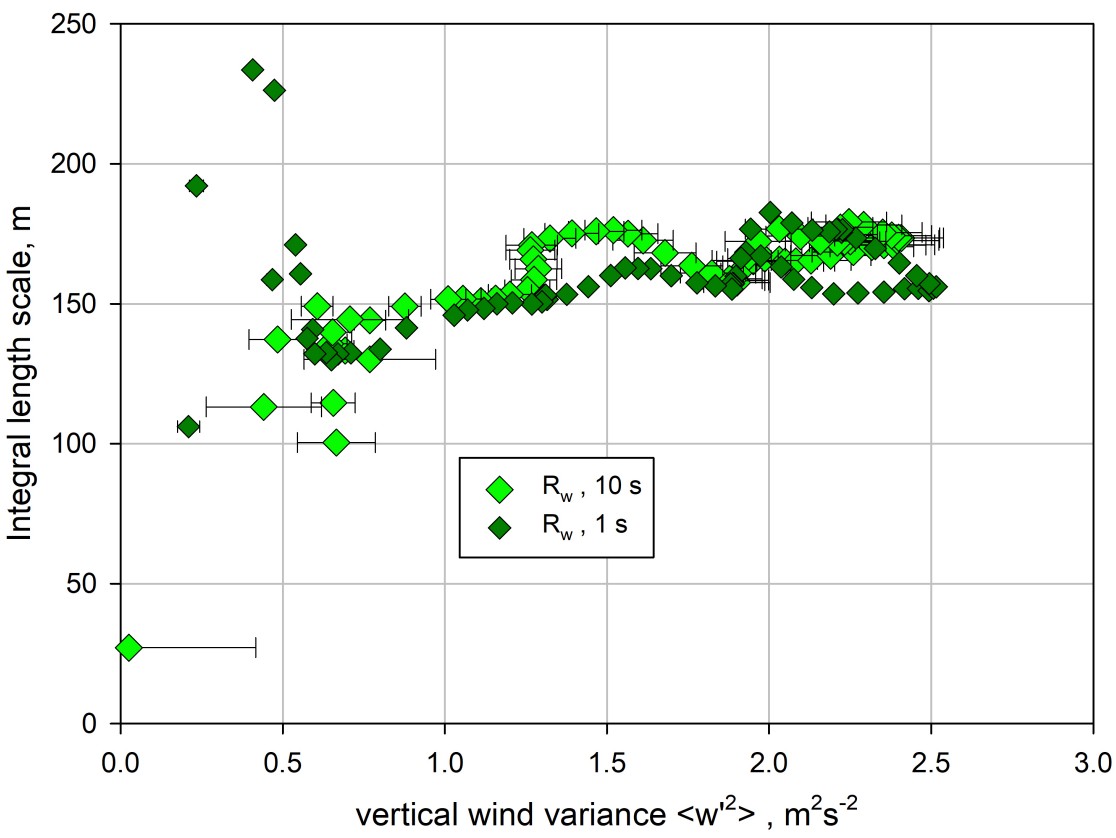

**Figure 20.** Dark green diamonds: $\mathcal{R}_w$ for 1-s data. Light green diamonds: $\mathcal{R}_w$ for 10-s data.

Further refinements are possible using the observed dependence of $\mathcal{R}_w$ on $\overline{w'^2}$, which is presented in Fig. 20. For $\overline{w'^2} > 0.5\,\mathrm{m\,s^{-1}}$, $\mathcal{R}_w$ seems to be fairly constant but it is uncertain in the IL. It is important to study whether our results for $\mathcal{R}_w$ and its small height dependence are universal so that our results can also be applied to other measurements or whether it is necessary to investigate more detailed similarity relationships for $\mathcal{R}_w$ in order to derive a more general parameterization of $\epsilon$.

## 5.2 Profiles of molecular destruction rates

Using high-resolution profile observations of the water-vapor mixing ratio $m$ and the potential temperature $\theta$, the equivalent methodology can be applied to derive profiles of the ACF coefficient and the molecular destruction rates for these two scalars.



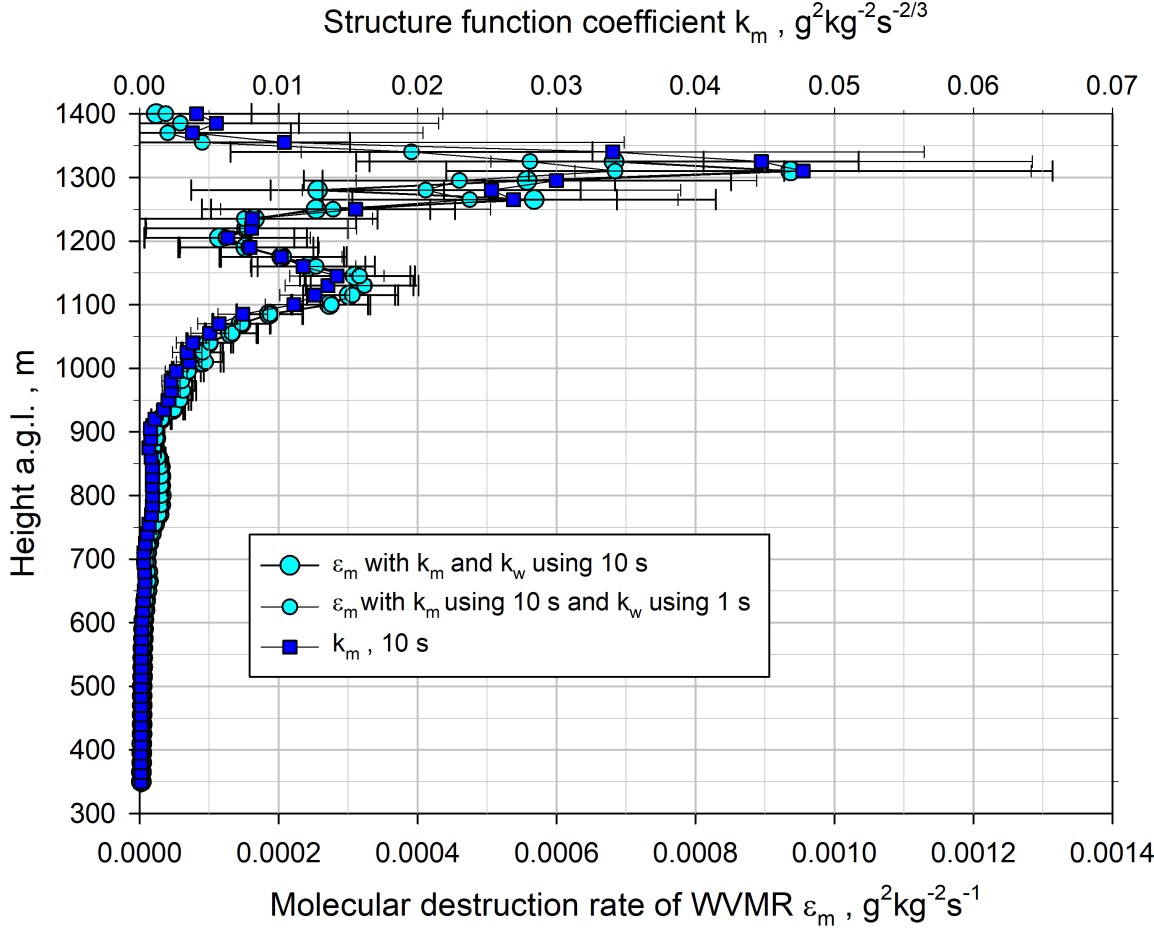

**Figure 21.** Small and large cyan circles: Profiles of $\epsilon_m$ derived with 1-s and 10-s resolutions for $k_w$, respectively. Large blue squares: Profile of $k_m$ with 10-s resolution.

### 5.2.1 Molecular destruction of mixing ratio variance

The derivation of the profiles of the ACF coefficient $k_m$ and the mixing ratio molecular destruction rate $\epsilon_m$ (see Eq. 20) is

presented in Fig. 21. We present only the $k_m$ profile for 10 s because it agreed very well with the 1-s profile in the ML but the

1-s data became unstable and partly negative in the IL. For the derivation of $\epsilon_m$, we used two options, either the $k_w$ profile for 1 s

or 10 s. Both resulting profiles are included in Fig. 21. Similar to $\overline{m'^2}$, both $k_m$ and $\epsilon_m$ maintain the double peak structure in the

CBL and reach a maximum in the IL very close to $z_i$. At the maximum, $k_m \simeq 0.05 \, \mathrm{g^2 kg^{-2} s^{-2/3}}$ and $\epsilon_m \simeq 7 \cdot 10^{-4} \, \mathrm{g^2 kg^{-2} s^{-1}}$

considering the more robust results from $k_w$ using the 1-s data. For a parameterization of $\epsilon_m$ it is very interesting to relate it

to both $\overline{m'^2}$ and $\overline{w'^2}$ using Eq. 21. The results are presented in Fig. 22. For this comparison we used all the data and a mean



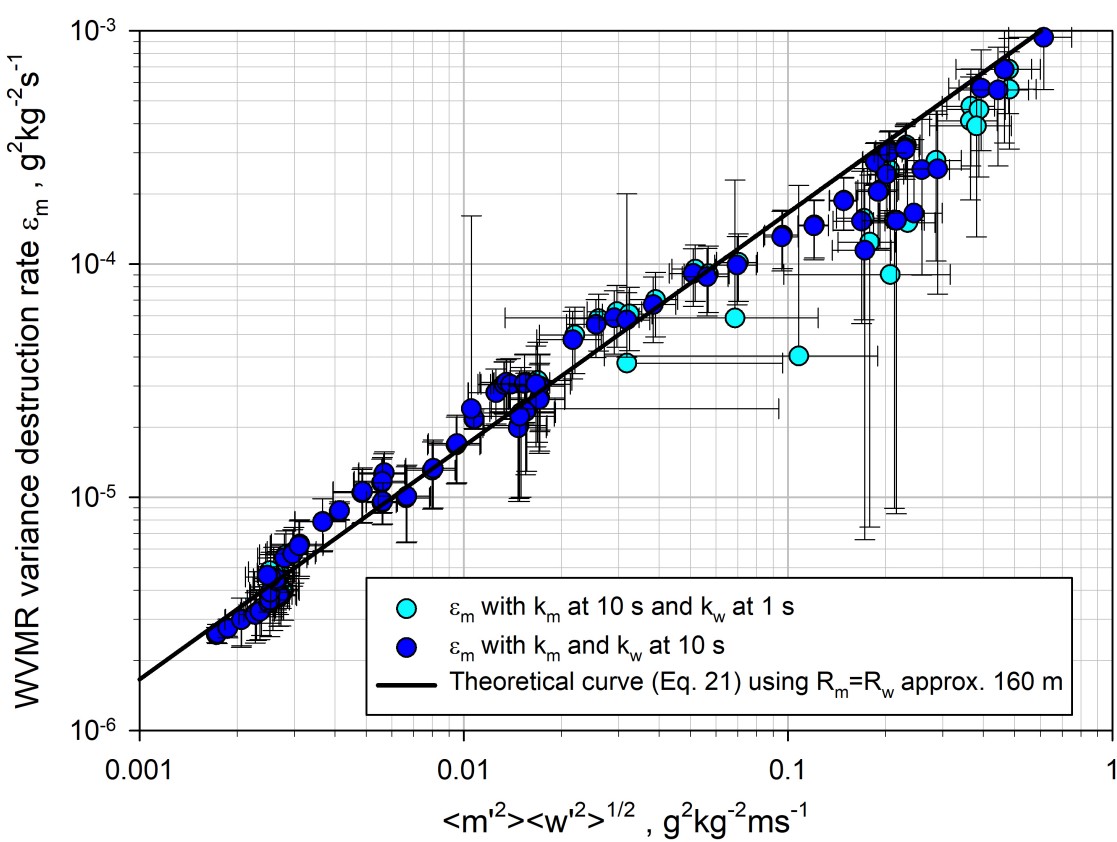

**Figure 22.** Cyan circles: $\epsilon_m$ derived with $k_w$ at 1-s resolution. Blue circles: $\epsilon_m$ derived with $k_w$ at 10-s resolution. Fit using Eq. 21 combining all data and using a mean spatial integral length scales of $\mathcal{R}_m = \mathcal{R}_w \simeq 160\,\mathrm{m}$.





integral length scale of $\mathcal{R}_m = \mathcal{R}_w \simeq 160\,\text{m}$ (see also Fig. 17) and achieved:

$$\epsilon_m \simeq \frac{4}{15}\frac{1}{160\,\text{m}}\overline{m'^2}\sqrt{\overline{w'^2}} \qquad , \tag{26}$$

which agrees well with our data over nearly three orders of magnitude. Most of the variability between theory and observation can be explained by the noise error bars. The data points of the 1-s data that deviate larger from the theoretical curve (e.g.,

$\overline{m'^2} \simeq 0.1\,\text{g}^2\text{kg}^{-2}$ and $\epsilon_m \simeq 4\cdot 10^{-5}\,\text{g}^2\text{kg}^{-2}\text{s}^{-1}$) belong to the top of the IL where the derivation of $\epsilon_m$ became particularly uncertain. The agreement of our observations with Eq. 21 indicates that a parameterization of $\epsilon_m$ is possible with the variances $\overline{m'^2}$ and $\overline{w'^2}$. Obviously, this kind of plot provides also the opportunity to estimate a kind of mean spatial integral scale between $\mathcal{R}_m$ and $\mathcal{R}_w$ by determining its slope.

### 5.2.2 Molecular destruction of potential temperature variance

The profiles of the ACF coefficient $k_\theta$ and the potential temperature destruction rate $\epsilon_\theta$ (see Eqs. 22 and 23) are presented in Fig. 23. As for $\epsilon_m$, $\epsilon_\theta$ was determined either using the $k_w$ profile for 1 s or 10 s, respectively. Similar to $\overline{\theta'^2}$, both $k_\theta$ and $\epsilon_\theta$ reach a maximum in the IL very close to $z_i$. At the maximum, $\epsilon_\theta \simeq 1.6\cdot 10^{-3}\,\text{K}^2\text{s}^{-1}$ and $k_\theta \simeq 0.09\,\text{K}^2\text{s}^{-2/3}$. Also here, there seems to be a second maximum below $z_i$ at approx. 1050 m. For a parameterization of $\epsilon_\theta$, we relate it to both $\overline{\theta'^2}$ and $\sqrt{\overline{w'^2}}$ (see Eq. 23). The results are presented in Fig. 24 with the theoretical function from Eq. 23. For this fit we used all the data and

approximated the spatial integral scales in the CBL by $\mathcal{R}_w \simeq 160\,\text{m}$ and $\mathcal{R}_\theta \simeq 200\,\text{m}$. Thus, we estimated

$$\epsilon_\theta \simeq \frac{4}{15}\frac{1}{(160\,\text{m})^{1/3}(200\,\text{m})^{2/3}}\overline{\theta'^2}\sqrt{\overline{w'^2}} \simeq \frac{4}{15}\frac{1}{186\,\text{m}}\overline{\theta'^2}\sqrt{\overline{w'^2}} \qquad , \tag{27}$$

which agrees reasonably well with our data. Almost all deviations between the theoretical curve and the observations can be explained by the large noise error bars.

## 6 Discussion

In this work, we used high-resolution time series of $w'$, $m'$, and $\theta'$ for profiling integral scales, variances as well as $\epsilon$, $\epsilon_m$, and $\epsilon_\theta$ in the CBL. We developed a technique to identify the suitable range of lags for fitting the theoretical shape of the ACFs to the data in order to consider the effects of temporal-spatial averaging/filtering effects using 1-s or 10-s data on the resulting profiles of turbulent quantities. We did not correct our measurements further with respect to filtering effects, as the corresponding loss is generally not larger than 10-15 %, as demonstrated and elaborated in Lothon et al. (2006, 2009). We

applied all our derivations in the temporal and spatial spaces because this handling of the data using the theoretical shape of the transversal ACF is equivalent to spectral analyses so that we avoid additional processing steps. We also calculated spectra to study the consistency and plausibility of our temporal statistics.

According to its definition, $\mathcal{T}_i$ is a kind of average for the typical duration of a coherent fluctuation in the inertial subrange or a way to characterize the corresponding horizontal size of a turbulent eddy according to $\mathcal{R}_i \simeq V\,\mathcal{T}_i$ assuming that the Taylor

hypotheses is applicable to these scales. Lothon et al. (2006) demonstrated that a length scale $L$ or an outer scale of turbulence

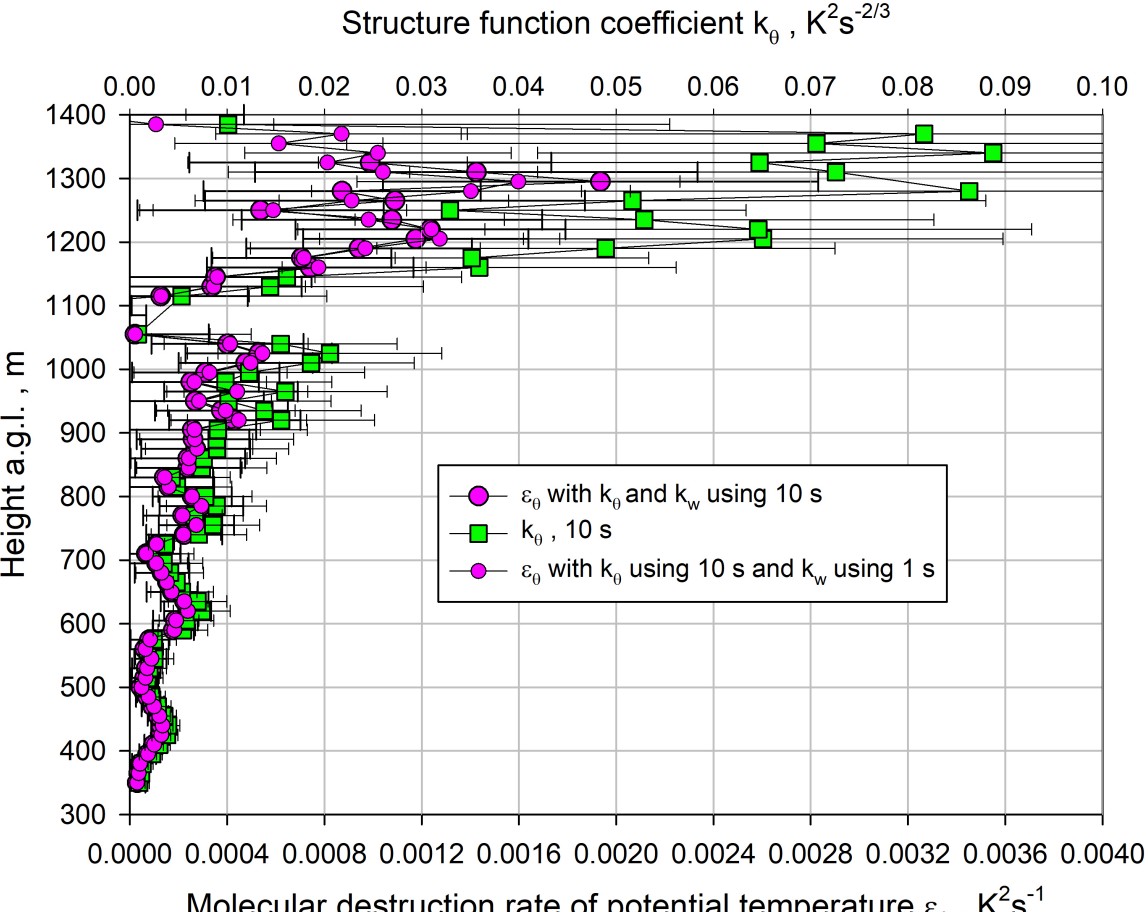

**Figure 23.** Pink circles: Profiles of $\epsilon_\theta$ derived with $k_\theta$ at 10 s resolution and $k_w$ at 1 s and 10 s resolutions, respectively. Green squares: Corresponding profiles of the ACF coefficients $k_\theta$ with 10 s resolution.

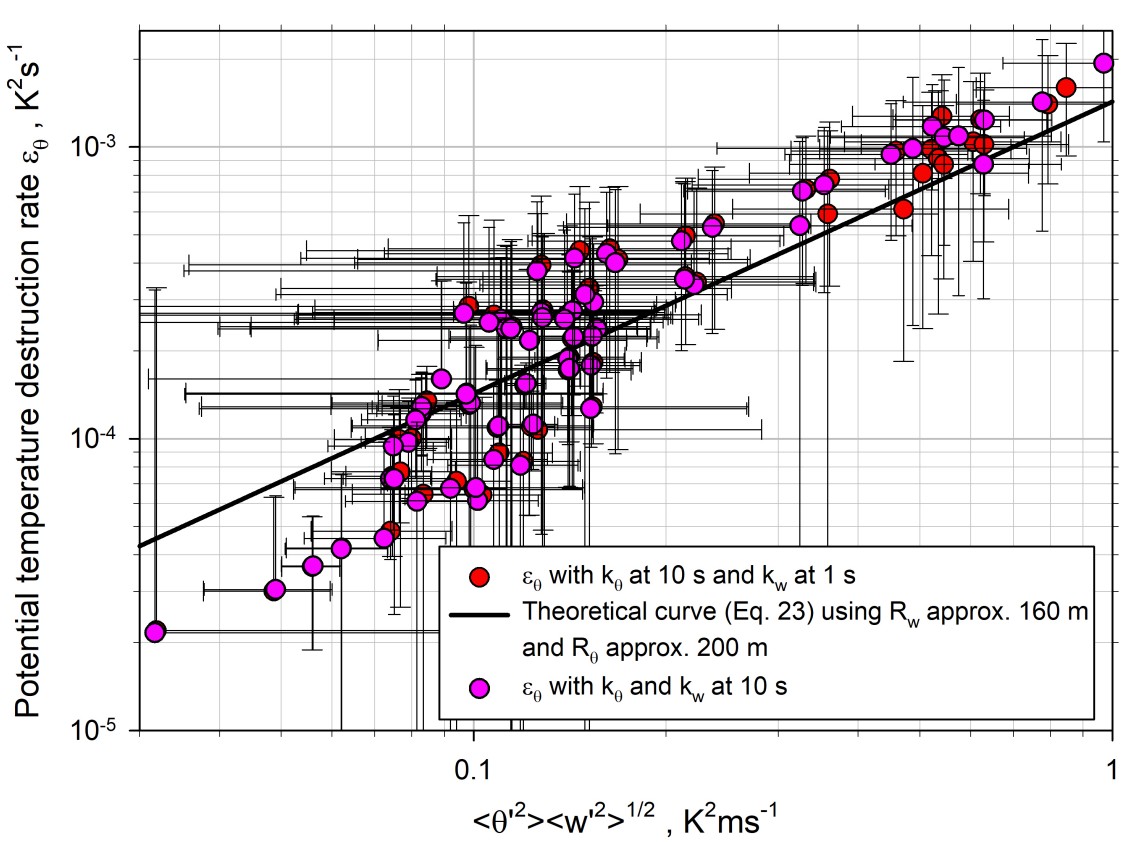

**Figure 24.** Red circles: $\epsilon_\theta$. Solid black line: Fit using Eq. 23 combining all data and using mean spatial integral length scales of $\mathcal{R}_w \simeq \mathcal{R}_m \simeq$ 120 m. Hollow red circles: Extrapolation.





can be defined where the inertial subrange become visible in the data, which resulted in $L_i \approx 2.7\,\mathcal{R}_i$. However, this concept only works, if the inertial subrange is not overwhelmed by mesoscale and microscale circulations so that its onset in the power spectra is highly variable. Related to these effects, it is essential to separate turbulent from micro- and mesoscale fluctuations in order to determine $\mathcal{T}_i$ correctly. The approximation to integrate the measured ACF to its first zero crossing is not the best approach and leads to an overestimation of $\mathcal{T}_i$ and $\mathcal{R}_i$ (compare, e.g., the measured and fitted ACFs on Fig. 7 at 545 m). It is fundamental to determine the range of the suitable lags first, derive $k_i$ from this, and then use Eq. 17 to derive $\mathcal{T}_i$ (Wulfmeyer et al., 2016). Eq. 17 provided rather robust results even at small variances and large noise levels (see Figs. 8, 12, and 16). For this procedure, it is not necessary to use a temporal resolution $\Delta t$ of the data to "resolve" the inertial subrange but it is required that a reasonable range of samples is present to realize an accurate fit of the data to the theoretical shape of the ACF. As the zero crossing $\tau_{0,i}$ of the extrapolated ACF is $5/2\,\mathcal{T}_i$ (Wulfmeyer et al. (2016)), $\Delta t << \tau_{0,i}$. This was the case for the 1-s and the 10-s data here because we found $\tau_{0,i} \simeq 5/2\,\mathcal{T}_i \simeq 50\,\mathrm{s} < 10\,\mathrm{s}$. This explains the good agreement of the profiles of temporal and spatial integral scales, the variances, the TKE dissipation, and the molecular destruction rates for both 1-s and 10-s resolutions. Thus, we could also estimate $L_i$ with $\tau_{0,i}$ or $\mathcal{T}_i$ because obviously $L_i \approx 2.5\,\mathcal{R}_i \simeq 2.5\,V\,\mathcal{T}_i \simeq V\,\tau_0$.

We determined $\mathcal{T}_w \simeq 20\,\mathrm{s}$ in the ML, which corresponds to $\mathcal{R}_w \simeq 160\,\mathrm{m} \simeq 0.13\,z_i$ (see Fig. 17), with an indication of a slight decrease towards the IL. Previous measurements of $\mathcal{T}_w$ and $\mathcal{R}_w$ were reported in Lenschow et al. (2000); Lothon et al. (2006). In the case studied in Lenschow et al. (2000), $\mathcal{T}_w \simeq 60\,\mathrm{s}$ resulting in $\mathcal{R}_w \simeq 180\,\mathrm{m} \simeq 0.12\,z_i$ with a slight increase towards the IL. In spite of this agreement, we need to evaluate more data consistently because Lenschow et al. (2000) used a linear fit to the ACF to derive $\mathcal{T}_w$ and $\overline{w'^2}$, which could have led to an overestimation of $\mathcal{T}_w$ and an underestimation of $\overline{w'^2}$. In Lothon et al. (2006); Lenschow et al. (2012) a number of cases were analyzed but here an integration of the ACF to the first zero crossing was used likely likely leading to an more severe overestimation of $\mathcal{T}_w$. This can be the reason that Lothon et al. (2006); Lenschow et al. (2012) found $\mathcal{R}_w \simeq 0.3\,z_i$ whereas we found $\mathcal{R}_w \simeq 0.13\,z_i$, which is a significant reduction. In any case, for future operational analyses of turbulence data it seems to be safe to estimate the upper limit of the number of suitable lags ($lag_{max}$) for fitting the ACF with its theoretical shape in the inertial subrange according to $lag_{max} < 2\,\mathcal{T}_w/\Delta t \simeq 0.25\,z_i/(V\,\Delta t) < \tau_{0,w}/\Delta t$. It is necessary to substantiate this relationship with large data sets from observatories such as LAFO (Späth et al., 2023).

With respect to $\mathcal{T}_m$, some results were presented in Lenschow et al. (2000); Wulfmeyer et al. (2016); Osman et al. (2018). Whereas we derived $\mathcal{T}_m \simeq 15\,\mathrm{s} \leq \mathcal{T}_w$ in the ML with a tendency to increase towards to the IL, $\mathcal{T}_m \simeq 50 - 100\,\mathrm{s}$ in these publications with a tendency to decrease towards the IL. It is essential to collect more data in combination with a consistent data processing as introduced in this work to get further insight in the statistics of $\mathcal{T}_m$ because similar to the discussion of $\mathcal{T}_w$, different integration schemes and numbers of lags were used likely resulting in an overestimation of $\mathcal{T}_m$. To the best of our knowledge, $\mathcal{T}_\theta$ was only studied in our work as well as in Behrendt et al. (2015); Wulfmeyer et al. (2016); Behrendt et al. (2020) using data from the same campaign but using a different number of lags, which explains that we specified now a smaller $\mathcal{T}_\theta \simeq 20\,\mathrm{s}$. The tendency of an increase of $\mathcal{T}_\theta$ towards the lower ML needs more investigation in the future.

Our profile of vertical velocity variance reaches a maximum at $\simeq 0.4\,z_i$ with a value of $\overline{w'^2} \simeq 2.5\,\mathrm{m^2 s^{-2}}$ and decreases to $\simeq 0.2\,\mathrm{m^2 s^{-2}}$ in the IL. This location of the maximum is similar to the observations in Lenschow et al. (2000, 2012); however, this maximum variance is significantly stronger, as for all cases reported in Lenschow et al. (2012) ($< 1.5\,\mathrm{m^2 s^{-2}}$).





This deviation can be partly due to the refined choice of lags for the fit of the ACF. The reduction of variance towards the IL rather consistent with the previously analyzed cases. In the CBL, in contrast to $\overline{w'^2}$, the profiles of $\overline{m'^2}$ and $\overline{\theta'^2}$ must peak in the IL because this is region of largest variables of $m$ and $\theta$. As shown in Lenschow et al. (2000); Wulfmeyer et al. (2010); Turner et al. (2014b); Muppa et al. (2016); Osman et al. (2018), the range of the peak values is large and our result ($\simeq 1\,\mathrm{g^2kg^{-2}}$)

is located well within the range of previous reported values ($0.4 - 4.4\,\mathrm{g^2kg^{-2}}$). It is not the subject of this work to study the relationship of the variance peaks to driving variables (for water-vapor mixing ratio see Wulfmeyer et al. (2016); Osman et al. (2019)). Interesting is the strong non-linear increase of $\overline{m'^2}$ in the ML towards the IL, which contains information about the turbulent properties of the CBL and will be investigated in future studies. According to our data, this increase is considerably less for $\overline{\theta'^2}$. In the IL, we derived a peak value of $\overline{\theta'^2} \simeq 1.1\,\mathrm{K^2}$, which is slightly larger than another reported in Behrendt et al.

(2015, 2020) and amounted $0.4\,\mathrm{K^2}$. We detected other peaks of the variance of $\overline{m'^2}$ at $1140\,\mathrm{m}$ and of $\overline{\theta'^2}$ at $1050\,\mathrm{m}$, which are unexpected and need also further investigations in the future.

Fundamental for the derivation of TKE dissipation of molecular destruction rates is the relationship between the variances and the coefficients $k_i$. Using Eq. 17, these read

$$var_{atm,i} = \left(\frac{5}{2}\mathcal{T}_i\right)^{2/3} k_i,\tag{28}$$

which agrees very well with our observations (not shown).

TKE dissipation $\epsilon$ can be derived by different techniques such as using the ACF (Davies et al., 2004), the structure function (Banakh et al., 2017), the power spectrum (Lothon et al., 2009; O'Connor et al., 2010; Lenschow et al., 2012; Bodini et al., 2018), and the Doppler spectral width (Doviak and Zrnić, 1993). The Doppler spectral width method is not very common anymore because most of the commercially available DLs do not store the full Doppler spectrum and the broadening of the

spectrum is not only due to turbulence but also due to several different effects such as wind shear. In contrast, the ACF, the structure function, and the power spectrum methods are straightforward and relatively easy to implement. Moreover, all these techniques are physically equivalent. As pointed out above, we prefer the use of the ACF because we avoid the introduction of further data processing steps and the noise introduced by the fast Fourier transform. Also, the noise error propagation is simple and can be automated along with the determination of $\epsilon$ itself.

The determination of $\epsilon$ by the power spectrum approach was described in O'Connor et al. (2010); Lothon et al. (2009); Lenschow et al. (2012); Bodini et al. (2018):

$$\epsilon \simeq 2\pi \left(\frac{2}{3\,a_k}\right)^{3/2} \frac{\left(\overline{w'^2}\right)^{3/2}}{L_{N,w}}\tag{29}$$

where $a_k \approx 0.55$ is the Kolmogorov constant and $L_{N,w}$ is a length scale of the large eddies. Comparison with Eq. 19 yields

$$L_{N,w} \simeq 5\pi \left(\frac{2}{3\,a_k}\right)^{3/2} R_w \simeq 21\,R_w \qquad,\tag{30}$$

which may be used for future derivations of $\epsilon$ with power spectra of ACFs. However, this relation seems to be inconsistent with the derivation of Lothon et al. (2006) (see above), which states $L_w \simeq 2.7\,\mathcal{R}_w$ (see above). It is currently not clear where this large deviation comes from but it may be the case that the definitions of $L$ and $L_N$ are not the same.





The magnitude of the $\epsilon$ profile that we observed agree quite well with those reported in previous studies including its reduction towards the IL (Lothon et al., 2009; O'Connor et al., 2010; Lenschow et al., 2012). Equations 29 and 19 can be used as parameterizations of $\epsilon$. For the turbulence parameterization in the CBL, TKE is mainly determined by $\overline{w'^2}$ so that for TPs such as MYNN a parameterization of $\epsilon$ by $\overline{w'^2}$ in the form of $\epsilon \propto e^{3/2}/l \simeq \overline{w'^2}/l$ is reasonable and should be tested in more detail in the future. This corresponds to the parameterizations of $\epsilon$ chosen in Nakanishi and Niino (2009); Olson et al. (2019), where $l$ is the mixing length scale (see Eq. 2). Obviously, Eq. 19 can not only be used to determine $\epsilon$ but also the missing coefficient $B_1$ coefficient there. Also, the profiling of $\epsilon$ can be applied for comparisons with LES and for TKE budget analyses (see Eq. 1 and Moeng and Wyngaard (1989); Sullivan and Patton (2011)).

In contrast to TKE dissipation measurements, we are not aware of the use of any remote sensing efforts to determine values or even profiles of the molecular destruction rates for temperature and water vapor variances. Both Wulfmeyer et al. (2016) and Osman et al. (2018) described the first attempts to study the coefficients of the ACFs; however, in this work we managed to derive profiles of these destruction rates quantitatively. Previous measurements were only possible with in-situ sensors. Corresponding measurements were reported in Caughey and Palmer (1979) and Lenschow et al. (1980) and yielded results of $\epsilon_m$ in the range of $10^{-4} - 10^{-3}\,\mathrm{g^2 kg^{-2} s^{-1}}$ around $z_i$. Our remote sensing approach yielded $\epsilon_m \simeq 8 \cdot 10^{-4}\,\mathrm{g^2 kg^{-2} s^{-1}}$ in the IL, which is very consistent with previous in-situ measurements. However, with our combined remote sensing capability it is possible to determine profiles of these molecular destruction rates and other important turbulence quantities in the CBL routinely so that a large database of results can be collected and evaluated.

It is interesting to relate the ACF coefficients, variances, and the molecular destruction rates in more detail. We have demonstrated how this can be done using a combination of spectral and ACF analyses, as done above for $\epsilon$. In the inertial subrange of the one dimensional spectra of mixing ratio it is known that

$$\epsilon_m \simeq \frac{(2\,\pi)^{2/3}}{\beta} \epsilon^{1/3} \overline{m'^2} L_m^{-2/3} \tag{31}$$

where $\beta \simeq 0.82$ is a spectral constant (Lenschow et al., 1980). A corresponding equation holds for $\epsilon_\theta$. This relationship indicates that in theory the molecular destruction rates are proportional to their variances. Furthermore, inserting Eq. 29 and assuming that the turbulent length scales for vertical wind, humidity, and temperature are similar, we find

$$\epsilon_m \simeq \frac{2\,\pi}{\beta} \sqrt{\frac{2}{3\,a_k}} \overline{m'^2} \frac{\sqrt{\overline{w'^2}}}{L_{N,m}} \tag{32}$$

$$\epsilon_\theta \simeq \frac{2\,\pi}{\beta} \sqrt{\frac{2}{3\,a_k}} \overline{\theta'^2} \frac{\sqrt{\overline{w'^2}}}{L_{N,\theta}} \tag{33}$$

Of course, as studies of spectra and ACFs are equivalent, these equations are consistent with our relationships derived in Eqs. 18-23. Comparing Eqns. 32 and 33 yields

$$L_{N,m,\theta} \simeq 32 \overline{R}_{m,\theta} \quad , \tag{34}$$

which is similar to Eq. 30 and where $\overline{R}_{m,\theta}$ is a kind of mean spatial integral scale between $w'$ and $m'$ or $\theta'$.

Since our method allows the measurement of vertical profiles of $k_m$, $k_\theta$, $\epsilon_m$, and $\epsilon_\theta$, these data can be used for more extensive comparisons with LES, variance budget analyses, and for the development of parameterizations of molecular destruction rates.





The latter is becoming more important, as for the current and the next generation of TPs, 2nd and 3rd order closures are under investigation (e.g., Olson et al. (2019)). The parameterization of $\epsilon$ is shown in Eq. 25 and Fig. 19. We achieved very convincing agreement between the derivation of $\epsilon$ in dependence of $\overline{w'^2}$ and their theoretical relationship. The deviations between $\epsilon$ in dependence of $\overline{w'^2}$ can be almost entirely explained by the noise error propagations. This also holds for the theoretical relationships between $\overline{m'^2}\sqrt{\overline{w'^2}}$ and $\epsilon_m$ (see Fig. 22 and Eq. 26) as well as between $\overline{\theta'^2}\sqrt{\overline{w'^2}}$ and $\epsilon_\theta$ (see Fig. 24 and Eq. 27).

Also in these cases, the deviation between the theoretical curves and the observations can be explained by the larger noise error bars. We expect that further improvements of there relationships can be achieved, if the noise of the WVDIAL and the TRRL measurements is reduced and the observed height dependence of the spatial integral scales are considered.

In the future, our WVDIAL measurements should be improved with respect to SNR to achieve better performance. This is now possible because recent updates to that lidar system have resulted in an average power of the laser transmitter of up to

10 W (Späth et al., 2016). Also for the TRRL measurements, significantly better performance is possible, as demonstrated in Lange et al. (2019). Furthermore, DLs should be used that provide a better performance in the IL both for the observation of $w'$ and its turbulence statistics as well as for the derivation of horizontal wind profiles. These DLs are already commercially available. Furthermore, for operational profiling of TKE and TKE dissipation, we recommend the operation of two closely collocated DLs, one in a continuous vertically staring mode and the other one in a six-direction staring mode, as demonstrated

in Bonin et al. (2017). Using the latter configuration, not only horizontal wind profiles can be measured with high temporal resolution including the potential of additional noise suppression methods for optimizing the SNR in the IL, but additional data products such as TKE and the momentum flux profiles can be provided (Späth et al., 2023).

## 7 Conclusions

In this work, the transverse temporal ACFs were used to derive vertical profiles of TKE dissipation $\epsilon$ as well as the molecular

destruction rates of mixing ratio and potential temperature $\epsilon_m$ and $\epsilon_\theta$, respectively. The prerequisite for deriving representative values is the applicability of Taylor's hypothesis of frozen turbulence for the spatial and temporal scales in the inertial subrange, which we assumed to be applicable due to the quasi-stationary behavior of the CBL. The molecular destruction rates were derived by combining measurements of the profiles of the ACF coefficients in the inertial subrange. The data were provided by a combination of three high-resolution active remote sensing systems, a Doppler lidar (DL), a water-vapor differential

absorption lidar (WVDIAL), and a temperature rotational Raman lidar (TRRL). These systems were collocated at one site during the HOPE campaign (Macke et al., 2017).

For this purpose, we applied the methodology proposed in Wulfmeyer et al. (2016). We showed that our approach is equivalent to the use of the transverse spectra of the fluctuations of the vertical wind $w'$, mixing ratio $m'$, and potential temperature $\theta'$; however, our direct use of ACFs does not require additional data processing steps such as applying Fourier transforms to

the data. Also, the propagation of noise errors is straightforward and was included in the analyses and interpretation of our results. In particular, the synergy achieved by using this array of active remote sensing systems enables us to identify and to choose the correct range of lags in the inertial subrange to derive the results for all these quantities (TKE dissipation and the





molecular destruction rates of temperature and moisture variances), as these were consistent with the expected shape of the ACFs. Another important result was that we demonstrated a methodology to derive consistent profiles of turbulent variables in

dependence of their spatial and temporal resolutions (1 s and 10 s), as long as several lags are located in the inertial subrange, which can be studied by the temporal and spatial integral length scales. In order to compare and the evaluate corresponding data sets from different sites, all tools for the derivation of turbulent variables should be harmonized and made available for the scientific community. Therefore, we will start soon to make this software, which is currently written in IDL, available of software repositories such as Github.

A weakly convective case was selected from the HOPE dataset and derived profiles of temporal and integral scales as well as of variances. Several relationships between $z_i$, the integral scales, and the other length scale of turbulence were derived. For instance, we found that $\mathcal{R}_w$, $\mathcal{R}_m$, and $\mathcal{R}_\theta \simeq 160$ m are similar at least in the ML so that our results indicate that $\mathcal{R}_i \approx .0.13\, z_i$. Further evaluations of similar data sets with harmonized processing tools and steps are necessary to confirm whether this relationship is universal.

We found a maximum of $\epsilon \simeq 0.01$ m$^2$s$^{-3}$ at approx. $0.4\, z_i$ rolling off to small values of $\epsilon \simeq 1 \cdot 10^{-3}$ m$^2$s$^{-3}$ in the IL. We showed that the ACF coefficient $k_w \propto \overline{w'^2}$ and $\epsilon \propto \left(\overline{w'^2}\right)^{3/2}$ including an estimation of the slope between these variables. This resulted in a proposed parameterization of $\epsilon$, which can be applied in the TKE budget equation for higher-order parameterizations of CBL turbulence.

We also showed that $k_m \propto \overline{m'^2}$, $k_\theta \propto \overline{\theta'^2}$, $\epsilon_m \propto \overline{m'^2}\sqrt{\overline{w'^2}}$, and $\epsilon_\theta \propto \overline{\theta'^2}\sqrt{\overline{w'^2}}$. All these profiles have peaks in the IL so that

the shapes of $k_m$ and $k_\theta$ differ from that of $k_w$. This also explains the differences in the profiles of $\epsilon_m$ and $\epsilon_\theta$ with respect to $\epsilon$. The profiles of $\epsilon_m$ and $\epsilon_\theta$ rise from quite small values in the ML to maxima in the IL. In our case, these maxima amount to $\epsilon_m \simeq 7 \cdot 10^{-4}$ g$^2$kg$^{-2}$s$^{-1}$ for mixing ratio and $\epsilon_\theta \simeq 1.6 \cdot 10^{-3}$ K$^2$s$^{-1}$ for potential temperature in the IL.

This combination of measurements has been realized during the field campaigns HOPE and LAFE (Wulfmeyer et al., 2018) as well as at the LAFO site Späth et al. (2023). However, the methodology presented in this work can also be applied to larger

data sets such as from the ARM SGP site in the US, the DWD Meteorological Observatory (MOL) in Lindenberg, Germany, the Payerne observatory of Meteo Swiss at for vertical wind and water vapor. Such long-term data are essential to characterize turbulence profiles in dependence of meteorological conditions and to gain insight how well the results hold over a wide range of situations. With respect to water vapor variances, this idea was demonstrated in Turner et al. (2014b); Osman et al. (2019).

The long-term goal should be to provide routine analyses of diurnal cycles of turbulence profiles in different climate regions

for confirming the universality of scaling or to refine them by characterizations of wind shear, the strength of the inversion layer, and other potential scaling variables. A corresponding setup of instrumentation was proposed for the GEWEX Land Atmosphere Feedback Observatories (GLAFOs, Wulfmeyer et al. (2020)) so that these turbulence studies will also be a backbone of the proposed GLAFO sites. Their results can be applied more extensively for turbulence theory, comparisons with LES, turbulence parameterizations as well as TKE and variance budget analyses.



*Code and data availability.* The HOPE data used in this work are available from the Institute of Physics and Meteorology (IPM), University of Hohenheim. The IDL codes to derive the variables can also be provided by IPM.

*Author contributions.* V. Wulfmeyer: Manucript writing, data analyses, turbulence theory, methodology, and discussion; C. Senff: turbulence methodology, data analysis, review, editing; F. Späth: WVDIAL analysis and methodology; A. Behrendt, D. Lange: TTRL data analysis and methodology; R.M. Banta: turbulence theory, methodology, editing; W.A. Brewer, A. Wieser: DL lidar data analysis and methodology; D.D. Turner: methodology, review, editing.

*Competing interests.* The authors declare no competing interests.

*Acknowledgements.* This study was supported by the Federal Ministry of Education and Research (BMBF) program High Definition Clouds and Precipitation (HD(CP)$^2$), grant 89243019SSC000034 from the Department of Energy Atmospheric System Research (ASR) program, NCAR, and NOAA Atmospheric Science for Renewable Energy Program. We appreciate the support of Dr. Norbert Kalthoff from KIT, Karlsruhe, in setting up and operating the Doppler lidar and the radio sounding systems.





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
