# Peer review of "Profiling the Molecular Destruction Rates of Temperature and Humidity as well as the Turbulent Kinetic Energy Dissipation in the Convective Boundary Layer"

_Atmospheric Measurement Techniques, 2023_

## Author Response (AR2)

**Reviewer 1:**

The paper is well written by experienced scientists and documents the present (state-of-the-art) status of lidar remote sensing of turbulence properties in the boundary layer.

I have only several minor points.

What is the meaning of the molecular destruction rate? What is the relevance and importance of this quantity? This needs to be explained in the necessary detail. In the literature, the notation 'molecular destruction rate' only appears in connection with Wulfmeyer articles.

**Response: We appreciate very much the positive and very constructive review.**

**We agree and added the following text in the manuscript (see also response to Reviewer2):**

**L4 in abstract: "These are fundamental loss terms in the TKE as well as temperature and mixing ratio variance equations."**

**We will replace par. 2 in section 1 (l. 31-42) by this paragraph and add another reference:**

**"In a turbulent flow, eddy diameters span a range of length scales (Tennekes and Lumley 1972). For a turbulent boundary layer, the largest eddies, which have the largest velocity fluctuations and thus the largest contributions to the velocity variances and the TKE, generally scale with the depth of that layer. These large eddies break down into smaller and smaller eddies. Although the resulting small eddies contribute less to the variances and have much less energy, they are important because when they get small enough, the fluctuations are damped by molecular viscosity in such a way that they become the major sink for TKE in the budget equation. We will refer to this sink as the molecular destruction of variances or dissipation of TKE. The sources of turbulence thus occur at the largest turbulent scales, but the sink of the turbulence is at the smallest scales. Whether TKE is increasing or decreasing depends on the balance between the sources and the sink, so establishing the magnitude of the sink is a key to proper modeling of turbulent mixing effects.**

**A common approach to turbulence parameterization (TP) in mesoscale numerical weather prediction (NWP) models is to solve budget equations for TKE and other scalar variances, presented later in this paper. This procedure requires predictive equations for those variances, including temperature (or potential temperature) and water-vapor specific humidity. As is the case for TKE, the generation of scalar variance occurs at larger turbulence scales, but the major sink is due to molecular damping or "destruction" at the smallest scales. These rates must be parameterized in mesoscale NWP models."**

**In section 3.1, we will replace the text from L169-175 as follows:**

**"Here, $\varepsilon_m$ and $\varepsilon_\theta$ are the molecular destruction rates of $\overline{m'^2}$ and $\overline{\theta'^2}$, respectively. Our objective here is to directly measure profiles of $\varepsilon_m$ and $\varepsilon_\theta$. Their quantification becomes more interesting as higher-order turbulence closure schemes are under development where the temperature and mixing-ratio variances budgets have to be studied in great**

detail including all loss terms. Examples are the Mellor–Yamada–Nakanishi–Niino (MYNN)-Eddy 180 diffusivity mass flux (EDMF) scheme (Nakanishi and Niino, 2009; Olson et al., 2019), and the Cloud Layers Unified By Binormals (CLUBB) scheme (Golaz et al., 2002; Huang et al., 2022). Furthermore, the study of variance budgets is required for the parameterization of sub-grid clouds in mesoscale models (Van Weverberg et al., 2016). In LES and DNS, the molecular destruction rates are not parameterized but it is assumed that these are resolved or negligible. Thus, a comparison of their simulations and our measurements can be used to study the sub-grid scale closure of these models."

This explanation of 'molecular destruction rate' is needed in the introduction and briefly also in the abstract.

**Response: See our answer above. For instance, we will add a corresponding sentence in L4: "These are fundamental loss terms in the TKE as well as temperature and mixing ratio variance equations."**

Line 36: MYNN TP, MYNN-EDMF: Please explain!

**Response: Mellor–Yamada–Nakanishi–Niino (MYNN) turbulence parameterization (TP), Mellor–Yamada–Nakanishi–Niino (MYNN) - Eddy Diffusivity-Mass Flux (EDMF) scheme. Will be added.**

Line 36-37: destruction rate of temperature and moisture variances? Please explain!

**Response: Now explained in more detail (see above).**

Line 73: You mention: A breakthrough regarding temperature measurements by using TRRL was published in 2015 and 2019, however, the HOPE campaign and the only one case you discuss here is from 2013. This is confusing. Why do you mention techniques and methods that were not available in 2013?

**Response: The data published in Hammann et al. 2015 were also from the HOPE campaign. It just took some time for the publication. Therefore, this series of references is reasonable. Lange et al. 2019 demonstrated that the TRRL can be incorporated in a transportable box but the principles of operation were the same as in Hammann et al. 2015. This will be added in the text.**

Page 9, figure 1: Please explain VAD.

**Response: Velocity azimuth display (VAD). Will be added.**

The sub-sub-section 4.2.1 is rather long, pages 11-28. Can we have a more balanced set of sections? Other main sections (such as Sect 5) have only 5-6 pages.

**We assume that the reviewer is referring to section 4.2.2. Yes, this can be adapted and we will do so by adding more subsections for the vertical wind, mixing-ration, and potential temperature variances, respectively.**

Please mention Eq. (5) in the caption of Figure 3.

**Response: Will be added.**

Improve Figure 4 caption.

**Response: Thank you, this will be corrected.**

It would be better to plot the profiles with large symbols first, before plotting the profiles with small symbols. Then the profiles with small symbols are better visible.

**Response: Great idea, we applied this to all plots (see attachment for the improved plots).**

Figure 9: Mention Eq. (6) in the caption.

**Response: Will be added.**

Figure 14: Mention Eq. (7) in the caption.

**Response: Will be added.**

In the beginning of Section 5, a short introduction to the given presentations would be fine! What you are going to present and why, ...motivation?

**Response: We would like to omit this extension because we think that this section is self-explanatory. Here, we determine the profiles of TKE dissipation and molecular destruction rates as motivated in sections 1 and 3.1.**

Figure 18: Mention Eq. (18) ? ...or (19)? in the caption.

**Response: Yes, this will be added.**

Figure 21: Mention Eq. (20) in the caption.

**Response: Will be added, too.**

**Reviewer 2:**

The authors aptly recognize the importance of measuring the TKE dissipation rates and variance budgets towards validating/improving the current CBL turbulence parameterization schemes. The presentation of the analysis methodology using ACFs to derive spatial and temporal integral length scales, variances, TKE dissipation rate, and "molecular destruction rates" for temperature and moisture is thoroughly articulated. The authors' commitment to creating an analysis script suite for the BL community to use is appreciable and would serve as a very useful resource to analyze large datasets in the future. I recommend publishing this article with the following (minor) details addressed for completeness and to improve the overall clarity of the article.

The importance of measuring TKE and TKE dissipation rates in the CBL and their implications for BL sub-grid scale turbulence and cloud parameterization in mesoscale models have long been established in the literature. However, section 1 lacks any clear description of molecular destruction rates of temperature and moisture. If such a description was provided in previously published articles by the authors, including such references would certainly help the readers. Secondly, a brief discussion on the implications of quantifying and parameterizing the molecular destruction rates on improving the sub-grid scale parameterizations in mesoscale models is warranted in section 1.

**Response:**

**We thank the reviewer for the positive review and the very constructive comments. We agree that more details on the role of the TKE dissipation and the molecular destruction rates of variances should be provided.**

**We will add the following information:**

**L4 in abstract: "These are fundamental loss terms in the TKE as well as temperature and mixing ratio variance equations."**

**We will replace par. 2 in section 1 (l. 31-42) by this paragraph and add another reference:**

**"In a turbulent flow, eddy diameters span a range of length scales (Tennekes and Lumley 1972). For a turbulent boundary layer, the largest eddies, which have the largest velocity fluctuations and thus the largest contributions to the velocity variances and the TKE, generally scale with the depth of that layer. These large eddies break down into smaller and smaller eddies. Although the resulting small eddies contribute less to the variances and have much less energy, they are important because when they get small enough, the fluctuations are damped by molecular viscosity in such a way that they become the major sink for TKE in the budget equation. We will refer to this sink as the molecular destruction of variances or dissipation of TKE. The sources of turbulence thus occur at the largest turbulent scales, but the sink of the turbulence is at the smallest scales. Whether TKE is increasing or decreasing depends on the balance between the sources and the sink, so**

establishing the magnitude of the sink is a key to proper modeling of turbulent mixing effects.

A common approach to turbulence parameterization (TP) in mesoscale numerical weather prediction (NWP) models is to solve budget equations for TKE and other scalar variances, presented later in this paper. This procedure requires predictive equations for those variances, including temperature (or potential temperature) and water-vapor specific humidity. As is the case for TKE, the generation of scalar variance occurs at larger turbulence scales, but the major sink is due to molecular damping or "destruction" at the smallest scales. These rates must be parameterized in mesoscale NWP models."

In section 3.1, we will replace the text from L169-175 as follows:

"Here, $\varepsilon_m$ and $\varepsilon_\theta$ are the molecular destruction rates of $\overline{m'^2}$ and $\overline{\theta'^2}$, respectively. Our objective here is to directly measure profiles of $\varepsilon_m$ and $\varepsilon_\theta$. Their quantification becomes more interesting as higher-order turbulence closure schemes are under development where the temperature and mixing-ratio variances budgets have to be studied in great detail including all loss terms. Examples are the Mellor–Yamada–Nakanishi–Niino (MYNN)-Eddy 180 diffusivity mass flux (EDMF) scheme (Nakanishi and Niino, 2009; Olson et al., 2019), and the Cloud Layers Unified By Binormals (CLUBB) scheme (Golaz et al., 2002; Huang et al., 2022). Furthermore, the study of variance budgets is required for the parameterization of sub-grid clouds in mesoscale models (Van Weverberg et al., 2016). In LES and DNS, the molecular destruction rates are not parameterized but it is assumed that these are resolved or negligible. Thus, a comparison of their simulations and our measurements can be used to study the sub-grid scale closure of these models."

The role of noise error bars is significant yet barely addressed in the article. Considering that the authors use the arguments related to noise error bars in justifying the deviations between the variances and turbulence parameters in their discussions (sections 5 and 6), the inclusion of a brief description of the noise error propagation procedure is warranted. Such a description could be either included as part of the main text (subsection in section 4) or as a standalone appendix section (similar to Wulfmeyer et al. (2016), Appendix b.4). Such a description is needed to clarify the high uncertainty in the estimates presented in Figure 24.

Response: We do not consider this necessary because exactly these equations are already published in Wulfmeyer et al. 2016 (Appendix b.4), as recognized by the reviewer. However, we added references to the derivation of the error bars in all corresponding captions.

The following are two minor deficiencies in the presented figures that need addressing (mostly for clarity):

3.1. In Figures 8, 12, and 19, the 1 s averaged data points are barely visible. These figures could be replotted to improve the clarity and visibility of all/ more data points and also differentiate the error bars more clearly.

**Response: We changed the figures accordingly (see attachment) and will adapt the captions.**

3.2. Figures 16, 17, 18, 19, 21, 22, 23, and 24: it is difficult to differentiate the error bars for potential temperature variance in the IL altitudes from the line connecting the integral time scale values. Perhaps change the color of the line connecting the integral time scale values to improve legibility.

**Response: We will change the figures accordingly (see attachment) and will adapt the captions.**